

# Time quasilattices in dissipative dynamical systems

**Felix Flicker[1,2]**

**1** Rudolph Peierls Centre for Theoretical Physics, University of Oxford, Department of
Physics, Clarendon Laboratory, Parks Road, Oxford, OX1 3PU, UK
**2** Department of Physics, University of California, Berkeley, California 94720 USA

flicker@physics.org

## Abstract

We establish the existence of 'time quasilattices' as stable trajectories in dissipative dynamical systems. These tilings of the time axis, with two unit cells of different durations, can be generated as cuts through a periodic lattice spanned by two orthogonal directions of time. We show that there are precisely two admissible time quasilattices, which we term the infinite Pell and Clapeyron words, reached by a generalization of the period-doubling cascade. Finite Pell and Clapeyron words of increasing length provide systematic periodic approximations to time quasilattices which can be verified experimentally. The results apply to all systems featuring the universal sequence of periodic windows. We provide examples of discrete-time maps, and periodically-driven continuous-time dynamical systems. We identify quantum many-body systems in which time quasilattices develop rigidity via the interaction of many degrees of freedom, thus constituting dissipative discrete 'time quasicrystals'.



# 1  Introduction

The spontaneous breaking of translation symmetry occurs whenever a crystal grows from a liquid. The result is a reduction of the continuous symmetry down to a discrete symmetry, the space group of the lattice. Recently it was asked whether the same process can occur in time. The name 'time crystals' was coined for hypothetical systems which spontaneously break time-translation symmetry in their ground states [1–3]. It later transpired that such a process is impossible [4–6], although a loophole left open the possibility of breaking the *discrete* time translation symmetry of periodically-driven systems down to a multiple of the period [7–10]. This led to physical implementations in both cold atoms and nitrogen vacancy defects in diamond [11, 12].

Concurrent with these developments, it was realized that the notion of space group symmetry can be extended to include time. These 'choreographic crystals' may feature a higher symmetry, when considering their constituent elements in both space and time, than is revealed by any instantaneous snapshot [13]. Additionally, symmetry operations have been identified in periodically-driven 'Floquet crystals'. Example symmetry operations include so-called time glides, combining a translation in time with a mirror in space [14].

Together these studies establish an understanding of periodicity and disorder on an equal footing in time and space. At first thought these cases seem to exhaust the possibilities for long-range order. Yet there exist long-range ordered objects which are neither periodic nor disordered: between these two extremes we find 'quasicrystals', atomic decorations of 'quasi-lattices' which are aperiodic tilings consisting of two or more unit cells. Despite lacking periodicity, they nevertheless feature a form of long-range order, which can be seen from the possibility of their construction as slices through higher-dimensional lattices [15–18].

In this paper we demonstrate the existence of *time quasilattices*: aperiodic tilings of the time axis using unit cells of two different durations. Despite lacking periodicity, they feature long-time order deriving from the fact that the sequence can be generated as a slice through a periodic two-dimensional lattice spanned by two orthogonal directions of time. We identify the time quasilattices as trajectories within nonlinear dynamical systems, coarse-grained to the scale of simply asking whether we are on the left $L$ or right $R$ of the system. The sequence of symbols $L$ and $R$ thus obtained matches the sequence of cells of a 1D quasilattice. We find that precisely two quasilattice sequences can grow as stable, attracting trajectories in nonlinear systems. We term these the Pell and Clapeyron quasilattices. We provide a systematic method by which to 'grow' these time quasilattices, showing that each finite-duration periodic approximation is also a stable, attracting trajectory, which provides a method of physically implementing the result. Additionally, we present a pedagogical introduction to relevant techniques employed in the field of symbolic dynamics. While well-known in the study of nonlinear

systems, these techniques provide a range of possibilities for extending ideas in the fields of time crystals, Floquet crystals, choreographic crystals, and quasicrystals, and it is our hope that in presenting them here further richness of results can be attained in these respective fields.

We provide examples of time quasilattices in a range of dissipative dynamical systems: the discrete-time logistic map, the continuous-time autonomous Rössler attractor, and the continuous-time periodically-driven forced Brusselator. Extending the crystal lattice analogy, we further identify time quasi*crystals*: systems in which the symmetry of a periodic driving is spontaneously broken to the symmetry of a time quasilattice, in which the stability is made rigid by the interactions between the macroscopic number of degrees of freedom of a quantum many-body state. We examine a number of recent experimental proposals concerning discrete time crystals in driven dissipative many-body systems [19–21], identifying that several additionally host time quasicrystals. We detail experimental signatures of these new states of matter.

This paper proceeds as follows. In Sections 2 and 3 we provide background on the subjects of quasilattices/quasicrystals, and symbolic dynamics, respectively. The aim is to develop the connections between the areas of study, and so we provide a solid introduction in each case. In Section 4 we present the bulk of our results, demonstrating that, of the ten classes of physically-relevant one-dimensional quasilattices, precisely two can exist as stable, attracting orbits in discrete-time nonlinear dynamical systems. In Section 5 we extend our results to continuous-time dynamical systems, providing routes to a physically-testable implementation. In Section 6 we explain the additional criteria which systems with the symmetries of time quasilattices must fulfill in order to constitute true states of matter – *time quasicrystals* – and identify these structures in recent experimental proposals. Finally in Section 7 we provide concluding remarks, and discuss the relationship between time quasilattices and periodic space-time systems.

## 2 Quasilattices and Quasicrystals

Quasilattices are aperiodic tilings consisting of two or more unit cells. Despite lacking periodicity, the placement of cells is not random: an $N$-dimensional quasilattice can be generated as a slice through a $2N$-dimensional periodic lattice [15–17]. Certain properties are more naturally expressed in terms of this higher-dimensional lattice. For example, two- or three-dimensional quasilattices feature discrete rotational symmetries in their diffraction patterns which are forbidden by the crystallographic restriction theorem, which states that only 2-, 3-, 4-, or 6-fold symmetries are allowed for periodic tilings in these dimensions. The symmetries demonstrated by quasilattices' diffraction patterns (5-fold, 8-fold, 10-fold, and 12-fold [22]) are nevertheless permitted to crystal lattices in the higher-dimensional space through which they were sliced [23–25]. Equivalently we can say that quasilattices are objects whose reciprocal-space dimension does not match their real-space dimension [16, 17].

Here we focus on the case of one-dimensional quasilattices, which can be generated as cuts through two-dimensional lattices. We reserve the name 'quasicrystal' for physical systems (quasilattice plus atomic basis) in dimensions two and higher. The phrase 'quasilattice' is used for the mathematical structure describing the physical system. A large number of quasicrystals has been grown artificially, and there have even been found two naturally-occurring examples, both in the same Siberian meteorite [26, 27].

In one dimension, it is a standard convention to refer to both the physical systems and their mathematical descriptions as quasilattices, disallowing the use of the term quasicrystal. This permits a precise definition of quasicrystals as those systems featuring diffraction patterns with symmetries forbidden by the crystallographic restriction theorem [15–17, 28, 29]. This

definition precludes the possibility of quasicrystals in one dimension, as rotations are not well defined. The only break which we make with this convention is in Section 6 in which we identify time quasilattices stabilized by many-body interactions: in order to emphasize that these states constitute an extension of the concept of time crystals to include quasilattice symmetry, we term them *time quasicrystals*, despite the fact that they exist in one dimension of time.

We define a 1D quasilattice to be an aperiodic tiling of a one-dimensional space with tiles of two different lengths, generated as a cut through a two-dimensional lattice. A necessary but not sufficient condition for an aperiodic tiling of two tiles to be a quasilattice is that each cell appears with precisely two spacings [28]. In Figure 1 we demonstrate the 'cut-and-project' quasilattice construction: we draw an irrationally-sloped line through the two-dimensional lattice, intersecting a vertex. For simplicity we take a square lattice with a unit cell length of one, but any regular lattice is acceptable. Drawing a second line parallel to the first which intersects the 2D lattice at the opposite vertex of the same unit cell, we project all vertices of the 2D lattice down onto the 1D lines whenever the vertices fall between the lines. If the gradient of the line is $\tan(\alpha)$, the spacings of the projected points will be either $\cos(\alpha)$ or $\sin(\alpha)$. The result is therefore two different unit cells tiling the line, and the sequence is guaranteed to be aperiodic by the irrationality of $\tan(\alpha)$.

If we are only interested in the sequence of cells which appears, rather than the cells' relative lengths, a simpler method is available, which we term the 'intersection method', explained in reference [28]. Starting with the same irrationally-sloped line, whenever the line intersects a vertical line of the 2D lattice we write one symbol, say $R$, and whenever the line intersects a horizontal line of the 2D lattice we write a second symbol, say $L$. The sequence of $R$s and $L$s will match that given by the cut-and-project method.

The best-known example of a 1D quasilattice is the infinite Fibonacci word ('word' here intuitively referring to a sequence of letters, $R$ and $L$). This can be generated either by cut-and-project or intersection, using a line whose gradient is the golden ratio $\varphi$:

$$\varphi = (\varphi - 1)^{-1} = \frac{1}{2}\left(1 + \sqrt{5}\right).$$

Consider again the case where the line intersects a vertex of the lattice. Such cases form a set of zero measure, but are instructive here [28]. From the first intersection of a lattice line after the (unique) intersection with a lattice vertex, the sequence of cells begins as follows:

$$RLR^2LRLR^2LR^2L\ldots$$

This is shown in Fig. 1.

The Fibonacci quasilattice can also be generated by so-called 'inflation rules':

$$R \to RL, \qquad L \to R, \tag{1}$$

which are applied to every symbol, starting from the left of the word, at every iteration:

$$
\begin{aligned}
R \to RL \to\ & RLR \\
\to\ & RLR^2L \\
\to\ & RLR^2LRLR \\
\to\ & RLR^2LRLR^2LR^2L \\
\to\ & RLR^2LRLR^2LR^2LRLR^2LRLR \\
\to\ & \ldots
\end{aligned}
$$

The lengths of these sequences are the Fibonacci numbers $F_n$, and hence they are known as Fibonacci words. The infinite Fibonacci word $F_\infty$ is the Fibonacci quasilattice [30].

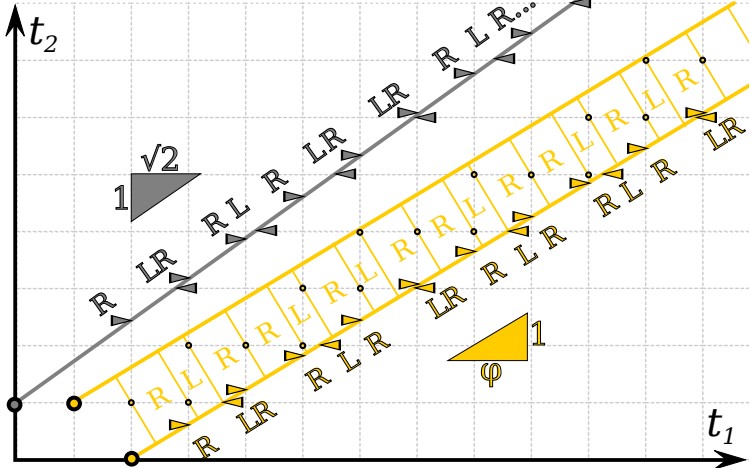

Figure 1: Different methods of creating a 1D quasilattice as a slice through a 2D lattice (as we are interested in time quasilattices, the 2D space is spanned by two orthogonal directions of time). Lower, in gold, is the Fibonacci quasilattice, made by taking a slice through the lattice at an angle given by the inverse of the golden ratio (or the ratio itself, which would interchange the cell labels). If a vertex of the 2D lattice falls between the parallel golden lines, it is projected onto them. The result is two cell lengths which we label $R$ and $L$. The sequence of long and short cells is the Fibonacci quasilattice (infinite Fibonacci word) $RLR^2LRLR^2L\ldots$. This is the 'cut-and-project' method. If we are only interested in the sequence of cells, we can simply write an $R$ whenever a vertical lattice line is intersected, and an $L$ when a horizontal line is intersected. The lower line is used to make these intersections, and the methods can be seen to agree. In silver this 'intersection' method is applied to a line with gradient $\frac{1}{\sqrt{2}}$, related to the silver ratio $1 + \sqrt{2}$. This is generating what we term the Pell quasilattice. Note that the intersection method in the Fibonacci case starts generating the correct sequence from the first intersection after the vertex, whereas the Pell case is offset along the line. This is because we have taken a non-canonical ordering of the Pell substitution rules (in the sense of [28]), for reasons to be explained later.

Note that, while each iteration of the inflation rules leads to extra cells growing on the right of the previous word, the growth mechanism is inherently nonlocal, requiring a substitution of every letter in the word simultaneously. The consistency of the leftmost string of each word is given by a discrete scale invariance implied by the inflation rules: considering the infinite Fibonacci word, while it can be described by the unit cells $R$ and $L$, it can equally-well be described by any inflation of these cells, for example $RL$ and $R$, $RLR$ and $RL$, *etc.* This property is shared with crystal lattices, which can be described by any integer multiple of their primitive unit cell.

While all quasilattices can be generated by cut-and-project, only some can be generated by inflation rules [17]. Conversely, inflation rules can generate objects which are not quasilattices, as we see in Section 3.5 when considering the period-doubling cascade. We can describe the Fibonacci inflation rules by a matrix

$$A = \begin{pmatrix} 1 & 1 \\ 1 & 0 \end{pmatrix}$$

such that

$$A \begin{pmatrix} R \\ L \end{pmatrix} = \begin{pmatrix} R + L \\ R \end{pmatrix}$$

and the $n^{\text{th}}$ term can be made by instead acting on the vector with $A^n$. The eigenvalues of $A$ are $\varphi$ and $\varphi^{-1}$. In general, quasilattices of two cell types can be described by matrices such as this, featuring non-negative integer entries, whose eigenvalues are Pisot-Vijayaraghavan numbers: quadratic irrationals $a + \sqrt{b}$ with rational $a$, $b$, such that $a + \sqrt{b} > 1$ and $0 < \left| a - \sqrt{b} \right| < 1$ [17, 28].

There are an infinite number of quasilattices which can be generated by the intersection method of Figure 1. They fall into equivalence classes. For example, shifting the intersecting line perpendicular to itself generates 'locally isomorphic' quasilattices, where any finite sequence of cells appearing in one appears in the others [15]. Loosely, these can be thought of as translations of one another. Others may be equivalent up to inflations, deflations, or translations of others [28]. While all $N$-dimensional quasilattices can be created through a cut-and-project from a $2N$-dimensional lattice, only a subset of these can also be generated through inflation rules [16, 17].

The recent work of reference [28] identifies that, in fact, only ten equivalence classes of 1D quasilattice exist which are truly physically relevant, in the sense that they relate to higher-dimensional counterparts through their identifying irrational numbers, which describe both the relative lengths of the cell types (volume, in general dimensions), and the relative frequency of the cells' appearances. In Table 1 we reproduce these quasilattices and their inflation rules from reference [28]. The physical significance of the ten classes is that their counterparts in two and three dimensions are precisely the quasicrystals which have been grown in the lab, or discovered as naturally-occurring materials [24, 25, 28]. This suggests their suitability for study with an eye to physical implementation. A theoretical argument due to Levitov underpins the experimental observation that all known two- and three-dimensional quasicrystals feature 5- 8- 10- or 12-fold symmetry [22]. The same argument explains why physical quasicrystals relate only to quadratic irrational numbers, whereas a more general class of similar objects can be constructed mathematically [18].

By way of example, consider again the Fibonacci quasilattice. This has cell lengths related by the golden ratio $\varphi$. The length of Fibonacci word $n$ is the $n^{\text{th}}$ Fibonacci number $F_n$, and we refer to the corresponding word with the same symbol $F_n$. The ratio of lengths of successive Fibonacci words also tends to the golden ratio:

$$\varphi = \lim_{n \to \infty} \frac{F_n}{F_{n-1}}.$$

The Fibonacci quasilattice generalizes to the two-dimensional case of the Penrose tiling, shown in Figure 2. The Penrose tiling's diffraction pattern has ten-fold rotational symmetry, forbidden in two-dimensional crystal lattices but allowed in four or five-dimensional crystal lattices through which it can be considered a slice [15, 28]. The ratio of the two cells' areas, as well as their relative frequency of occurrence, is given by the golden ratio. The full significance of the relationship between the Fibonacci quasilattice and the Penrose tiling requires a consideration of Coxeter groups and Ammann decorations, explained in detail in references [28, 29].

## 3 Symbolic Dynamics

In this section we present a brief overview of relevant results in the field of symbolic dynamics. Aside from being necessary background to the subsequent sections, the hope is that a pedagogical introduction to the ideas will be useful to authors working in the fields of time crystals, Floquet crystals, choreographic crystals, and quasicrystals. In Section 3.1 we define some basic terms in the study of general nonlinear dynamical systems. In Section 3.2 we present the basic ideas of symbolic dynamics. One technique, word lifting, is particularly important to the

Table 1: (After [28]). The ten equivalence classes of physically-relevant 1D quasilattices. The substitution matrix $A$, applied to the vector $(R, L)^T$, generates the substitutions $R \to \rho$, $L \to \lambda$ under addition. The 'slope' column indicates the slope of the line drawn through a square two-dimensional lattice to generate each sequence by either cut-and-project or intersection. The columns $\rho$ and $\lambda$ show the cell sequences generated from $R$ and $L$ respectively. Ref [28] identifies a canonical ordering of substituted cells; we do not follow this, as it grows the quasi-lattices symmetrically to the left and right of the starting point, whereas we will wish to refer to the right of the generated string as the future. See also the discussion in Section 4.1.

| Case | $A$ | eigenvalues | slope | $\rho$ | $\lambda$ |
|---|---|---|---|---|---|
| 1 | $\begin{pmatrix} 1 & 1 \\ 1 & 0 \end{pmatrix}$ | $\frac{1}{2}\left(1 \pm \sqrt{5}\right)$ | $\frac{1}{2}\left(1 \pm \sqrt{5}\right)$ | $RL$ | $R$ |
| 2a | $\begin{pmatrix} 1 & 1 \\ 2 & 1 \end{pmatrix}$ | $1 \pm \sqrt{2}$ | $\pm\sqrt{2}$ | $RL$ | $R^2L$ |
| 2b | $\begin{pmatrix} 0 & 1 \\ 1 & 2 \end{pmatrix}$ | " | $1 \pm \sqrt{2}$ | $L$ | $RL^2$ |
| 3a | $\begin{pmatrix} 1 & 2 \\ 1 & 3 \end{pmatrix}$ | $2 \pm \sqrt{3}$ | $\frac{1}{2}\left(1 \pm \sqrt{3}\right)$ | $RL^2$ | $RL^3$ |
| 3b | $\begin{pmatrix} 2 & 1 \\ 3 & 2 \end{pmatrix}$ | " | $\pm\sqrt{3}$ | $R^2L$ | $R^3L^2$ |
| 3c | $\begin{pmatrix} 1 & 1 \\ 2 & 3 \end{pmatrix}$ | " | $1 \pm \sqrt{3}$ | $RL$ | $R^2L^3$ |
| 4a | $\begin{pmatrix} 3 & 1 \\ 4 & 1 \end{pmatrix}$ | $2 \pm \sqrt{5}$ | $-1 \pm \sqrt{5}$ | $R^3L$ | $R^4L$ |
| 4b | $\begin{pmatrix} 2 & 1 \\ 5 & 2 \end{pmatrix}$ | " | $\pm\sqrt{5}$ | $R^2L$ | $R^5L^2$ |
| 4c | $\begin{pmatrix} 1 & 1 \\ 4 & 3 \end{pmatrix}$ | " | $1 \pm \sqrt{5}$ | $RL$ | $R^4L^3$ |
| 4d | $\begin{pmatrix} 0 & 1 \\ 1 & 4 \end{pmatrix}$ | " | $2 \pm \sqrt{5}$ | $L$ | $RL^4$ |

present work, and Section 3.3 is devoted to it. Section 3.4 presents the 'generalized composition rule', which is the key mathematical tool used to prove the existence of time quasilattices. The results are returned to frequently in later sections. Finally, in Section 3.5, we apply the ideas to the period-doubling cascade into chaos, both as an already well-understood example, and to provide a point of reference when explaining the generalization to the Pell and Clapeyron cascades in Section 4.

## 3.1 Nonlinear Dynamics Definitions

In this section we briefly define some concepts in the study of nonlinear dynamical systems and chaos which will be referred to later in the paper. Detailed introductions and more precise definitions can be found for example in references [30, 32, 33].

Dynamical systems are defined by their equations of motion, which take the general form

$$\dot{\mathbf{x}} = \mathbf{f}(\mathbf{x})$$

with $\mathbf{x}$ a vector describing the state at a given time. In this paper we will consider $\mathbf{x}$ to be

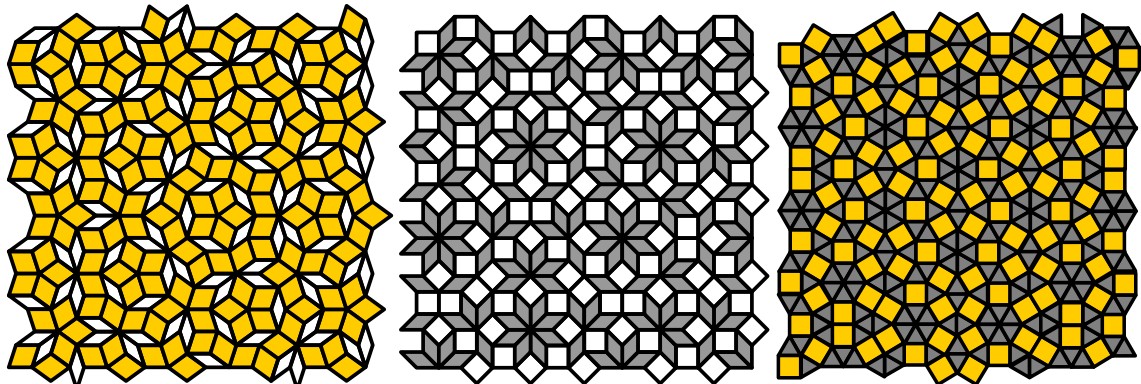

Figure 2: Sections of two-dimensional quasicrystals, after [18]. **Left**: the Penrose tiling. The ratio of the areas of the tiles is the golden ratio $\varphi$, as is the relative frequency of occurrence of the two tiles [31]. Approximate 5- or 10-fold symmetry can be seen at different points, and the diffraction pattern features true 10-fold symmetry. This is the two-dimensional generalization of the Fibonacci quasilattice, which also features cell lengths and relative frequencies in the golden ratio. **Middle**: the Ammann-Beenker tiling. The ratio of cell areas is $\sqrt{2}$. The tiles' relative frequencies lie in the silver ratio $1 + \sqrt{2}$. Approximate 8-fold symmetry can be seen at points, and the diffraction pattern is 8-fold symmetric [23]. This is the two-dimensional generalization of the Pell quasilattice, similarly related to the silver ratio. **Right**: The two-dimensional generalization of the Clapeyron quasilattice (which has cell lengths related by $2 + \sqrt{3}$). The diffraction pattern is 12-fold symmetric.

positions. If the equations are such that they can be rewritten as a Lagrangian

$$L(\mathbf{x}, \dot{\mathbf{x}}) = \frac{1}{2}\dot{\mathbf{x}}^2 - V(\mathbf{x})$$

the system has an associated Hamiltonian and conserves energy; otherwise it is dissipative. If time only enters the equations implicitly via $\mathbf{x}(t)$ the equations are said to be autonomous, and if time appears explicitly through some driving they are non-autonomous (driven). Non-autonomous equations can be rewritten in an autonomous form by introducing extra variables, as we will show in Section 5 when dealing with the forced Brusselator.

The continuous-time systems we consider in Section 5 feature 'attractors', regions which attract trajectories from within a wider 'basin of attraction' and to which trajectories converge at infinite time. A 'strange attractor' additionally has a fractal structure. A convenient tool for analyzing continuous-time dynamical systems is the Poincaré section, a slice through trajectories. This can be used to construct the Poincaré first-return map, which plots intersection $n + 1$ of a trajectory with the Poincaré section against intersection $n$, provided the trajectory passes through the section in the same direction.

We refer to trajectories in discrete-time maps as 'orbits', which need not be periodic (closed). When considering continuous-time systems we will always have in mind the relation to the discrete-time maps via the Poincaré first-return map, and so refer to orbits in these cases also. An orbit is 'Lyapunov stable' if all trajectories which start sufficiently close to it remain so for all time [32]. If a Lyapunov-stable orbit is also attracting, then it is said to be stable, although we often reiterate the attracting nature of stable orbits here. The Lyapunov stability of an orbit can be characterized by its Lyapunov exponents, which are calculated using a local linearization of the nonlinear map at a point in time. At a maximum the linear term vanishes. In the cases considered here this leads to 'superstability', which can be thought of either as a Lyapunov exponent of $-\infty$ or as trajectories converging onto the orbit at a faster-than-exponential rate [34]. There is a separate notion of 'structural stability', meaning the orbits

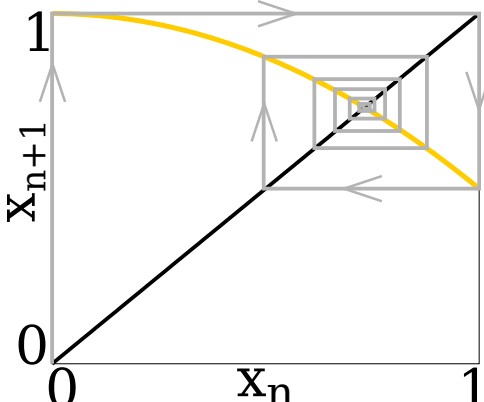 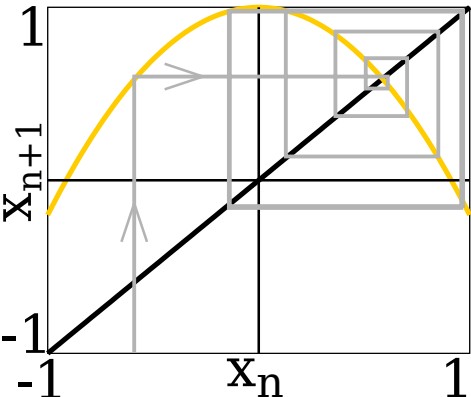

Figure 3: The logistic map $x_{n+1} = 1 - \mu x_n^2$. **Left**: for $\mu = 0.5$ the map features a single stable fixed point. This can be found by the 'cobwebbing' technique: draw a vertical line to the map, then a horizontal line to the diagonal $x_{n+1} = x_n$, then iterate. This feeds the output of one iteration into the input of the next, and so on. **Right**: for $\mu = 1.2$ the fixed point is unstable, but there exists a stable period-2 orbit onto which trajectories converge (again found through cobwebbing).

under consideration are unaffected by sufficiently small changes to the system parameters. All cases considered in this paper are structurally stable.

## 3.2 Symbolic Dynamics Background and Nomenclature

In this section we present a pedagogical introduction to some basic concepts in symbolic dynamics. The ideas are presented clearly in reference [34], an excellent resource. Other classic references include [30, 35–37].

In the majority of this paper we focus on the logistic map:

$$x_{n+1} = 1 - \mu x_n^2 \tag{2}$$

with real-valued $x \in [-1, 1]$. Note that this is not the standard writing of the map, but leads to a neater analysis in what follows. The map is shown in Fig. 3 for several choices of the parameter $\mu$, along with the corresponding stable, attracting orbits of the dynamics. The logistic map models a discrete-time dissipative dynamical system, with each iteration constituting a time step. Despite the map's simplicity, it features stable periodic orbits of all periods, as well as chaos [32, 35, 37]. The map was originally introduced as a simplified model of animal population dynamics [38]. Additionally, it falls into the same universality class as a wide range of continuous-time nonlinear dynamical systems. In Section 5 we demonstrate that the results found for the logistic map can be extended to continuous-time dynamical systems of both the autonomous (Rössler) and periodically-driven (forced Brusselator) type. These extensions provide testable predictions for physical systems, but the simplicity of the logistic map makes it invaluable in introducing and proving the relevant ideas.

Symbolic dynamics applied to the logistic map entails a coarse-graining to simply asking whether we are on the left, $L$ ($x < 0$), or right, $R$ ($x > 0$), of the maximum [35–37]. The $x_n$ in the iterations are found to an appropriate precision, and recorded. Once the desired number of iterations has been carried out, we assign a letter to each according to whether it is greater or smaller than zero. The central point $x = 0$ has a special significance, since its appearance in an orbit implies superstability: there is no linear expansion of the curve at its maximum, so the Lyapunov exponent is $-\infty$, and trajectories converge faster than exponentially onto

such orbits. We denote this point $C$. Iterations of the map can be carried out by the 'cobwebbing' technique: drawing successive vertical lines to the map and and horizontal lines to the diagonal line $f(x) = x$. If we consider the set of points of the diagonal line which are hit, and which side of the map they fall onto, we will find a sequence of letters taken from the set $\{R, L, C\}$. This sequence of letters is termed a word. When the word is periodic we write only the repeating part. We will be interested in finding 'admissible' words: those which describe stable, attracting, orbits.

The superstable period two sequence is described by the word $RCRC\ldots = (RC)^{\infty}$, from now on denoted $RC$. The word $LC$ is inadmissible, as there is no intersection of the line $f(x) = x$ in the range $-1 < x < 0$. The admissible words of the logistic map turn out to be universal, in the sense that the same words are the admissible words of any differentiable 1D maps. If the map is additionally unimodal, having a single maximum like the logistic map, the sequence in which the words develop as the parameter $\mu$ is increased is also universal [36]. It is called the 'universal sequence', and it provides a re-ordering of the set of integers (so that each integer appears once as the length of a stable orbit) [39]. Continuous-time dynamical systems fall into this same universality class provided they are sufficiently dissipative. Note that all sufficiently dissipative continuous-time strange attractors have differentiable 1D discrete maps as their Poincaré sections, and all differentiable 1D discrete-time maps can be related to continuous-time attractors [40]. In fact, dissipation may be too strong a requirement, as we discuss in Section 7 when considering non-dissipative Hamiltonian systems.

All orbits longer than two are described by words beginning $RL$, as the trajectory first goes to the largest $x$ it will ever hit (positive, so $R$), then the smallest $x$ it will hit (negative, so $L$). This provides a useful constraint when searching for quasilattices, as it specifies that the first cell generated by the inflation rule must be labeled $R$, and the second $L$. Simple algorithms exist for determining the admissibility of a given word, and therefore dynamical trajectories. Reference [34] gives an extensive explanation of various methods. One of these, the generalized composition rule, we explain and employ shortly.

Despite a great deal of overlap between the study of quasiperiodic dynamical systems and the study of quasilattices, the latter have not previously been identified as admissible trajectories in dynamical systems. A comment is necessary regarding the relation to quasiperiodic systems, which feature two or more incommensurate frequencies, and which are well-studied in the context of dynamical systems. Quasiperiodic systems could either be described as having an uncountably-infinite number of unit cells of different lengths, or as having one unit cell of infinite length, or neither. Quasilattices, on the other hand, have a finite number of unit cells, with most studies focussing on the case of two [41]. They therefore feature a minimum and maximum spacing between cells [16,17]. If we consider an irrationally-sloped line drawn through a 2D lattice, as in Fig. 1, the spacing of intersections of the line with the lattice is quasiperiodic, and may be infinitesimally small. The sequence of symbols generated, however, is a quasilattice.

References [30,42] consider a simple linear quasiperiodic system defined by the map

$$x_{n+1} = x_n + \alpha \tag{3}$$

on the interval $x \in [0, 1)$ (with $x = 0$ and $x = 1$ identified) for quadratic irrational $\alpha$. By dividing the domain into two partitions, a coarse-graining is defined which leads the quasiperiodic motion to spell the corresponding word describing the quasilattice. The lack of nonlinearity, however, makes these states 'marginally stable', meaning they have zero Lyapunov exponent, and can be destroyed with infinitesimal perturbations. This is the key difference with the quasilattice states we identify here, which maintain their sequences under such perturbations. Stability is key in the present study, as in the previous work on time crystals (a point we return to in Section 6) [7–10].

It should be noted that, in any chaotic regime, all periodic and aperiodic orbits appear but are unstable [37]. This includes both the quasilattice trajectories we seek, and their periodic approximations. In Anosov chaotic systems, obeying Smale's Axiom A, unstable orbits have both stable and unstable manifolds [43,44]. If a trajectory starts on an orbit's stable manifold, it stays close to the orbit for all time, although it is unstable to perturbations taking it off the manifold. 'Control of chaos' involves protocols to direct trajectories onto given stable manifolds, and to stabilize chosen orbits within chaotic systems [45–48]. These techniques require the evolution to be monitored, and tailored perturbations to be added based on both the particular system and the evolution of the trajectory. In the present work we seek to drive a dynamical system *purely periodically*, and to receive a response which spontaneously breaks the symmetry down to that of a quasilattice. In this way the work generalizes the results of previous work on time crystals, in which a periodic driving received a response breaking the symmetry to a periodic response of twice the period. For this reason we disallow symmetry-breaking perturbations on top of the driving. We discuss what can be termed true *time quasicrystals* in Section 6.

### 3.3 Word Lifting

Once a sequence is established to be admissible, it is necessary to identify the parameter settings which will allow its realization. In the logistic map of Eq (2), it is necessary to identify the parameter $\mu$. This process is known as 'word lifting' [34].

The logistic map is many-to-one, so its inverse is multivalued. In taking the inverse we have to specify which of the two branches to take, left or right. So define two inverse functions like so:

$$f_L^{-1}(x) \triangleq L(x) = -\mu^{-\frac{1}{2}}\sqrt{1-x}$$
$$f_R^{-1}(x) \triangleq R(x) = \mu^{-\frac{1}{2}}\sqrt{1-x},$$

where '$\triangleq$' indicates a definition. A period-$N$ orbit is defined by

$$f \circ f \circ f \circ \ldots f(x) = x$$

with $N$ nested functions. We will focus on superstable fixed points, which feature the points $x = 0$ and $f(0) = 1$. To invert the sequence of maps we have to specify which branch to take at each iteration. Take the example of the superstable period five sequence $RLRRC$:

$$f(f(f(f(f(0))))) = 0$$
$$\downarrow f(0) = 1$$
$$f(f(f(f(1)))) = 0$$
$$\downarrow$$
$$f(f(f(1))) = R(0)$$

and, inverting the other functions to form the original word,

$$1 = R \circ L \circ R \circ R(0)$$
$$1 = \mu^{-\frac{1}{2}}\sqrt{1 + \mu^{-\frac{1}{2}}\sqrt{1 - \mu^{-\frac{1}{2}}\sqrt{1 - \mu^{-\frac{1}{2}}\sqrt{1-0}}}}$$
$$\downarrow \times \mu$$
$$\mu_{n+1} = \sqrt{\mu_n + \sqrt{\mu_n - \sqrt{\mu_n - \sqrt{\mu_n}}}}.$$

The indices introduced in the final expression indicate an iterative expression with which to find $\mu$ numerically. In general, for the logistic map, each letter in the word simply dictates the corresponding $\pm$ in the sequence in the final expression. As $\mu$ must still be found numerically, it is simpler to define separate functions for $R$ and $L$, and iterate the expression

$$\mu_n = R_{\mu_{n-1}} \circ L_{\mu_{n-1}} \circ R_{\mu_{n-1}} \circ R_{\mu_{n-1}} (0),$$

where '$\circ$' indicates function composition. It is a testament to the power of the technique that, despite each nested function introducing an additional square root, we are able to apply word lifting to seventy-letter words without issue.

### 3.4 Maximal Sequences and the Generalized Composition Rule

In this section we define some terms and operations used later in the paper. We stick to common conventions, and refer the reader to the references for further explanation and proofs [32, 34, 39].

**Parity of Words**: the parity of a word $\Sigma$, $P(\Sigma)$, can be established by the following facts:

$$P(R) = 1$$
$$P(L) = -1$$
$$P(\Lambda\Sigma) = -P(\Lambda)P(\Sigma)$$

that is, the parity of the word counts the number of $R$s; a word with an odd number of $R$s is said to be odd, and, somewhat counter-intuitively, has parity $+1$.

**Order of Words**: letters are ordered $L < C < R$, which just corresponds to their ordering along the real line. Given two words

$$W_1 = W^*\sigma \dots$$
$$W_2 = W^*\tau \dots,$$

where $W^*$ is common to both words, the order of the words is as follows:

$$W^* \text{ even,} \begin{cases} \sigma > \tau & \to W_1 > W_2 \\ \sigma < \tau & \to W_1 < W_2 \end{cases}$$
$$W^* \text{ odd,} \begin{cases} \sigma > \tau & \to W_1 < W_2 \\ \sigma < \tau & \to W_1 > W_2. \end{cases}$$

Note that the order of two words matches the order in which they appear as $\mu$ is increased in the logistic map, or the equivalent of $\mu$ is increased in a general unimodal differentiable map [36, 39].

**Maximal Words**: the significance of maximality is that any superstable periodic orbit is described by a maximal word, and if any maximal word has its last letter substituted with a $C$ its orbit becomes superstable [37]. A word $\Sigma$ is maximal iff

$$\Sigma \geq S^k(\Sigma) \quad \forall k$$

where the shift operator $S^k$ removes the first $k$ letters of the word (shifts the symbols by $k$ to the left). If the word is of finite length, it is maximal if it is larger than all its subshifts [30].

**The generalized composition rule**: this is a method of generating maximal sequences by substituting letters into already known maximal sequences. Using the notation that $\Sigma|_C$ is the word $\Sigma$ with its final letter substituted with a $C$, given a maximal word $\Sigma$, the substitutions $R \to \rho$ and $L \to \lambda$ also yield a maximal word if the following are true:

1. $P(\lambda) = P(L), \quad P(\rho) = P(R)$

2. $\rho > \lambda$

3. $\rho|_C$ is maximal

4. $\rho\lambda|_C$ is maximal

5. $\rho\lambda^\infty$ is maximal.

We again refer to the references for the proof, but note that the first rule ensures that substitutions maintain the parity of the word, and the other rules ensure the substitution's maximality [34].

**The Periodic Window Theorem**: any parameter $\mu$ corresponding to a superstable orbit must be contained in a window of values $\mu$ of finite measure corresponding to stable (but not superstable) orbits [34,37]. This can be seen for the logistic map in Fig. 4. Note that while the window is guaranteed to exist and be of finite measure, its actual width is system-dependent, and cannot necessarily be simply determined. It is not even true, for example, that either longer words or later words in the universal sequence have smaller windows [34].

## 3.5 Application to the Period-Doubling Cascade

In order to demonstrate the use of these concepts, we consider the the period-doubling cascade route to chaos in the logistic map. The working in this section is well-understood, but it provides a useful reference when considering the Pell cascade in subsequent sections [32,34, 37,49,50]. Period doubling can be generated by repeated use of the substitutions

$$
\begin{aligned}
R &\to \rho = RL \\
L &\to \lambda = RR.
\end{aligned}
\tag{4}
$$

This is the simplest possible nontrivial substitution compatible with the generalized composition rule, as parity (number of $R$s) must be preserved, meaning $\rho$ must have an odd number, and $\lambda$ an even number, of $R$s. Applied to the initial symbol $R$ we have:

$$
\begin{aligned}
R \to &RL \\
\to &RLR^2 \\
\to &RLR^3LRL \\
\to &RLR^3LRLRLR^3LR^3 \\
\to &RLR^3LRLRLR^3LR^3LR^3LRLRLR^3LRLRL \\
\to &\dots
\end{aligned}
$$

The length of the word after $n$ iterations is $2^n$. Figure 4 shows the points cycled between once transients have died down, *i.e.* the points constituting the stable orbit, for each value of $\mu$ in the logistic map of Eq. (2). The plot is known as an orbit diagram [32]. Inspecting the letters of each word generated by Eq. (4), we find the sequences of points visited in the uppermost part of each periodic window (after the line crosses $x = 0$) in each period doubling in Fig. 4. Since the mapping in Eq. 4 obeys the criteria of the generalized composition rule, and since the starting term $R$ is maximal and admissible, each term is therefore maximal and admissible.

Some of the smaller periodic cycles are shown to the right of the diagram. For $\mu < \frac{3}{4}$ there is a single stable fixed point which is converged to for all starting conditions. While the value $x_\infty$ converged to depends on $\mu$, it always lies on the right of the map, so is described

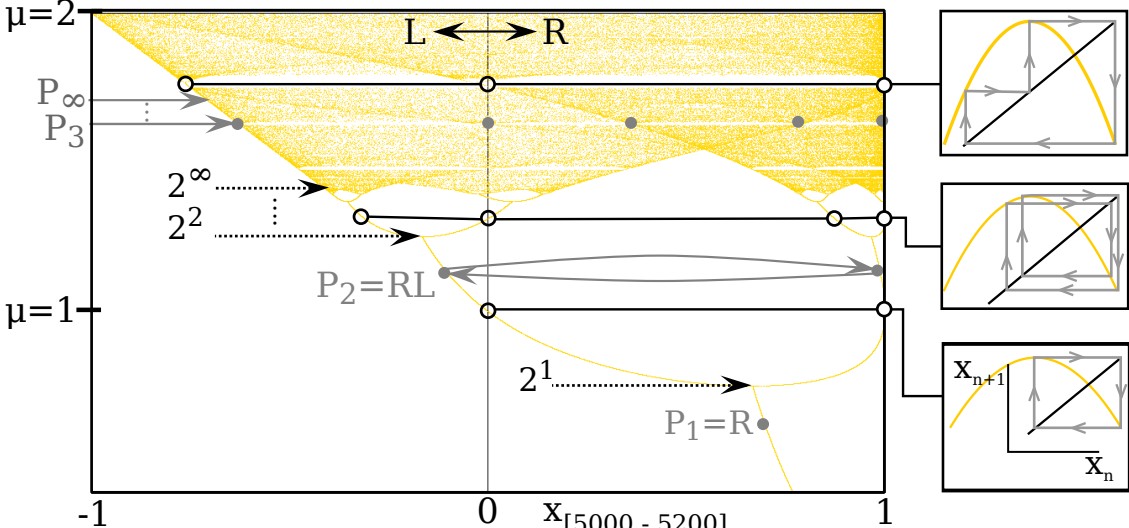

Figure 4: The orbit diagram of the logistic map of equation (2). The map is iterated for 5000 steps at each value of $\mu$, to allow stable orbits to converge, then 200 points are plotted. Points $x < 0$ are on the left of the map, and are labeled $L$ (labels $x = 0 : C$, $x > 0 : R$). Period doublings are marked with dashed lines. Chaos results above $2^\infty$ at $\mu \approx 1.4012$. The boxes on the right show the superstable orbits of period 2, 4, and 3 (bottom to top). Additionally, the grey solid arrows indicate the sequence of superstable orbits constituting the Pell cascade (Section 4.3). The Pell quasilattice $P_\infty$ is reached by $\mu \approx 1.6703$.

by the word $R$. At $\mu = \frac{3}{4}$ there is a period-doubling bifurcation to a stable 2-cycle (period 2 orbit) which can be found analytically. At $\mu = \frac{5}{4}$ this period 2 orbit becomes superstable, as it contains the point $x = 0$ (implying the other point, $x = 1$). This process then repeats an infinite number of times: with increasing $\mu$, a stable $2^n$ orbit described by the word $\Sigma R$ ($\Sigma L$) becomes superstable, $\Sigma C$, then stable $\Sigma L$ ($\Sigma R$), then a period-doubling bifurcation occurs to orbit length $2^{n+1}$ (note that the terminal letter alternates, hence the importance of the words' parities). An infinite number of such bifurcations then occurs in a period-doubling cascade, until at around $\mu \approx 1.42$ period $2^\infty$ is reached, and a chaotic regime is entered.

The periodic window theorem can be seen in action in Figure 4. The value $\mu = 1$ corresponds to a superstable period-2 orbit, since $f(x) = 1 - x^2$ iterates between $x = 0, 1$. Symbolically, the corresponding word is $RC$. There is a finite range of values $\frac{3}{4} < \mu < 1$ where the period-2 orbit $RR$ is stable ($R$ in our conventions), and a finite range $1 < \mu < \frac{5}{4}$ where the period-2 orbit $RL$ is stable. Equivalent results can be seen for the higher period-doubled orbits in the cascade.

It is interesting to note that the period-doubling cascade generated by the substitution rules of Eq. (4) shares many properties with the growth of a quasilattice: each new word contains the previous word as its leftmost string, and the final word in the period-doubling cascade will consist of an infinitely-long aperiodic string of two symbols with two spacings between each symbol (zero or one between $Rs$, one or three between $Ls$) [30]. These are necessary but not sufficient conditions for quasilattices. To check whether the infinitely period-doubled word is a quasilattice, we can examine its substitution matrix:

$$A = \begin{pmatrix} 1 & 1 \\ 2 & 0 \end{pmatrix}$$

which has eigenvalues 2 and -1. The requirement for a $2 \times 2$ substitution matrix to define a quasilattice is that its eigenvalues are Pisot-Vijayaraghavan numbers: one must be greater

than one, and the absolute magnitude of the other must be strictly less than one [16, 28]. The period-doubling sequence therefore fails on the grounds that the smaller eigenvalue has a magnitude of *exactly* one. This is why it does not appear in Table 1. While only an infinitesimal difference, it is the difference between rationality and irrationality. There can be no cut-and-project construction corresponding to the matrix $A$, since the intersection line would require rational slope 2, which implies a periodic sequence of cells, yet the inflation rules are generating an aperiodic sequence.

# 4 Growing Time Quasilattices

In this section we present our main results. Building on the background in the previous sections, we demonstrate the existence of 'time quasilattices': tilings of the time axis by two unit cells of different duration, in an aperiodic pattern which can be described as a slice through a two-dimensional tiling of a space spanned by two orthogonal time directions. We require time quasilattices to be both stable to perturbations, and to attract nearby trajectories in the phase space. We also require them to 'grow' in a systematic manner, by inflation rules.

   We will show that the quasilattice cells, labeled $R$ and $L$, correspond to the letters in admissible words in symbolic dynamics, and therefore the points visited at each iteration of a discrete-time differentiable unimodal map coarse-grained into two halves. We consider the specific example of the logistic map, but the results apply to the map's entire universality class, which contains all discrete-time differentiable 1D maps, as well as continuous-time dynamical systems which are sufficiently dissipative that their dynamics are well-represented by Poincaré sections which are approximately 1D. This point is returned to in Section 5 in which we consider physical implementations of the results.

   This section proceeds as follows. In Section 4.1 we identify that there are precisely two physically-relevant 1D quasilattices admissible as stable attracting orbits in nonlinear dynamical systems. We term these the infinite Pell word, and the infinite Clapeyron word. In Section 4.2 we prove the result rigorously using the tools of symbolic dynamics. Focussing on the simpler Pell word, in Section 4.3 we identify an analogue to the period-doubling cascade route to chaos, which we term the 'Pell cascade', which provides a systematic growth mechanism of the infinite Pell word by successive periodic approximations (finite Pell words). These provide a practical method of implementing time quasilattices in finite-duration experiments.

## 4.1 Admissible Time Quasilattices

Employing the tools explained in the background sections, proving the existence of time quasilattices reduces to the task of identifying quasilattice inflation rules of the form

$$R \to \rho, \quad L \to \lambda,$$

which obey the generalized composition criteria of Section 3.4.

   Checking the ten sets of inflation rules listed in Table 1 against the criteria of Section 3.4, it seems that there is only one match. The Fibonacci words of class 1 are ruled out, for example, as the corresponding inflation rules do not preserve the words' parities. The first admissible class is 2a, generated by the substitution rules

$$R \to \rho = RL, \quad L \to \lambda = RRL. \tag{5}$$

The corresponding substitution matrix

$$A = \begin{pmatrix} 1 & 1 \\ 2 & 1 \end{pmatrix}$$

has eigenvalues $1 \pm \sqrt{2}$: the silver ratio and its Galois conjugate. The sequence of words generated by the substitutions applied to $R$ is

$$
\begin{aligned}
R &\to RL \\
&\to RLR^2L \\
&\to RLR^2LRLRLR^2L \\
&\to RLR^2LRLRLR^2LRLR^2LRLR^2LRLRLR^2L \\
&\to \dots
\end{aligned}
$$

of lengths 1, 2, 5, 12, 29, 70, ... (sequence A000129 in OEIS [51]). These are known as the Pell numbers, $P_n$, after Euler's inaccurate attribution of their discovery to John Pell [52]. We refer to the words as 'Pell words', and denote both the Pell word and its length by $P_n$. The silver ratio is to the Pell words $P_n$ as the golden ratio is to the Fibonacci words $F_n$, *i.e.*

$$
1 + \sqrt{2} = \lim_{n \to \infty} \frac{P_n}{P_{n-1}}.
$$

The infinite Pell word $P_\infty$ we term the Pell quasilattice. This quasilattice was previously considered in reference [53].

Inspecting the inflation rules of the remaining nine quasilattice classes other than the Pell quasilattice, it appears at first that all others are incompatible with the generalized composition rules. However, some care has to be taken, since the inflation rules listed in Table 1 are ambivalent to the labels attached to each cell. For example, class 3a is listed as having substitution rules

$$
R \to RL^2, \qquad L \to RL^3,
$$

which do not preserve parity (number of $R$s). If we relabel the cells $R \leftrightarrow L$ we have

$$
L \to LR^2, \qquad R \to LR^3,
$$

which do preserve parity. Additionally, we are free to cyclically permute the letters in the substituted sequences $\rho$ and $\lambda$, as this just corresponds to translating the corresponding quasilattice by a finite number of symbols (time steps). Cyclic permutation is allowed provided that it preserves the topology of the quasilattice. This can be understood by attempting to re-order the Pell inflation rules as follows:

$$
R \to RL, \quad L \to LRR,
$$

which would lead to the words:

$$
R \to RL \to RL^2R^2 \to RL^2R^2LR^3LRL \to \dots
$$

The sequence is not growing a quasilattice, as it does not obey the necessary condition of having two cell types with two spacings between each cell. On the other hand, the topology-preserving re-ordering

$$
R \to RL, \quad L \to RLR
$$

leads to the words

$$
\begin{aligned}
R &\to RL \\
&\to RLRLR \\
&\to RLRLR^2LRLR^2L \\
&\to RLRLR^2LRLR^2LRLRLR^2LRLR^2LRLRLR \to \dots
\end{aligned}
$$

which obey the necessary condition, and can be seen to be leading to a translation of the usual Pell quasilattice (simply note that the distribution of $Rs$ and $Ls$ is tending to the silver ratio as before). This re-ordering fails to fulfill the criteria of Section 3.4, however.

Classes 1, 2b, 3b, 4b, and 4d are inadmissible in any combination, as the rules violate parity. Considering all possible cyclic permutations of the letters within $\rho$ and $\lambda$ in the remaining classes, ten can be re-arranged into admissible forms. However, a quick check of the inflated words shows that nine do not constitute quasilattices. The result is that there is only one additional quasilattice which can lead to stable attractive orbits in nonlinear dynamical systems. It is defined by the inflation rules of class 3a, adjusted to the following form:

$$R \to RLR^2, \quad L \to LR^2, \tag{6}$$

which lead to the substitutions

$$
\begin{aligned}
R \to &RLR^2 \\
\to &RLR^2 LR^3 LR^3 LR^2 \\
\to &RLR^2 LR^3 LR^3 LR^2 LR^3 LR^3 LR^3 LR^2 LR^3 LR^3 LR^3 \dots
\end{aligned}
$$

*i.e.* words of length 1, 4, 15, 56, 209, ... (OEIS A001353). Up to sign differences these are known as the 'Clapeyron numbers' (OEIS A125905) after appearing in a treatise on beam bending by Clapeyron [54]. We name the corresponding words the Clapeyron words $C_n$, with the infinite Clapeyron word $C_\infty$ being the Clapeyron quasilattice. The ratios of cell lengths of the Clapeyron words asymptotically approach the value

$$2 + \sqrt{3} = \lim_{n \to \infty} \frac{C_n}{C_{n-1}}.$$

The analyses presented throughout this paper apply equally well to the Clapeyron quasilattice as to the Pell quasilattice, although we focus on the simpler Pell case.

## 4.2 Proof of Maximality of the Pell Words

To prove all Pell words are maximal, it is sufficient to show that the Pell substitution rules of Equation (5) obey the generalized composition rules of Section 3.4. We restate and address criteria (1)-(5) of Section 3.4 individually.

**(1)** $P(\lambda) = -1, P(\rho) = 1$:

$$
\begin{aligned}
P(\rho) &= -P(R)P(L) = 1 \checkmark \\
P(\lambda) &= -P(R)P(RL) = P(R)P(R)P(L) = -1 \checkmark
\end{aligned}
$$

**(2)** $\rho > \lambda$:

The common word appearing as a leftmost string between $\rho$ and $\lambda$ is $W^* = R$, which is of odd parity. The next letter in $\rho$ is $L$, and that in $\lambda$ is $R$. Since $L < R$, and the common word is of odd parity, this implies that $\rho > \lambda \checkmark$

**(3)** $\rho|_C$ is maximal:

We must check that all finite shifts of the word $\Sigma = \rho|_C$, *i.e.* truncations of the word from the leftmost letter, are smaller than $\Sigma$ itself. The only shifted word is $\Sigma_1$:

$$
\begin{aligned}
\Sigma &= \rho|_C = RC \\
\Sigma_1 &= C.
\end{aligned}
$$

The common word $W^* = blank$, which contains an even number of $R$s (zero) and is therefore even. The next letter $R > C \therefore \Sigma > \Sigma_1$, so $\rho|_C$ is maximal ✓

**(4)** $\rho\lambda|_C$ is maximal:

$$\Sigma = \rho\lambda|_C = RLRRC$$

and, as before, we check all shifts of the word ($b$ is the blank word):

$$\Sigma_1 = LRRC, W^* = b, \text{even}, L < C \therefore \Sigma > \Sigma_1$$
$$\Sigma_2 = RRC, W^* = R, \text{odd}, L < R \therefore \Sigma > \Sigma_2$$
$$\Sigma_3 = RC, W^* = R, \text{odd}, L < C \therefore \Sigma > \Sigma_3$$
$$\Sigma_4 = C, W^* = b, \text{even}, R > C \therefore \Sigma > \Sigma_4$$

$$\therefore S^k\left(\rho\lambda|_C\right) < \rho\lambda|_C \ \forall k$$

and $\rho\lambda|_C$ is maximal ✓

**(5)** $\rho\lambda^\infty$ is maximal:

$$\Sigma = \rho\lambda^\infty = RL(RRL)^\infty$$

$$\Sigma_1 = L(RRL)^\infty, W^* = b, \text{even}, R > L \therefore \Sigma > \Sigma_1$$
$$\Sigma_2 = RRL(RRL)^\infty, W^* = R, \text{odd}, L < R \therefore \Sigma > \Sigma_2$$
$$\Sigma_3 = RL(RRL)^\infty = \Sigma$$

therefore

$$S^k(\rho\lambda^\infty) \leq \rho\lambda^\infty \ \forall k$$

and $\rho\lambda^\infty$ is maximal ✓

The Pell inflation rules therefore obey the generalized composition criteria. Since the word $R$ is maximal, and the Pell words are generated by application of the Pell inflation rules to $R$, all Pell words are therefore maximal, including the Pell quasilattice $P_\infty$. □

An identical analysis can be applied to the Clapeyron word inflation rules to prove their admissibility, and can be applied to all other quasilattices listed in Table 1 to see that these two are the only possible time quasilattices. In all cases the results can be verified by attempting to locate the corresponding values of $\mu$ using word-lifting: if the words are inadmissible, even a single iteration of $\mu_{n+1}(\mu_n)$ is likely to fail, whereas the value of $\mu$ converges for a wide range seed values even for the length 70 Pell word (requiring 69 nested square-root functions).

### 4.3 The Pell Cascade

The transition to chaos in the logistic map comes about through an infinite number of applications of the period-doubling substitutions $R \to RL$, $L \to RR$ applied to the symbol $R$. This is know as a period-doubling cascade, as each iteration leads to a word twice the length of its predecessor. The Pell quasilattice is generated through an infinite number of applications of the Pell inflation rules, Equation 5, to the symbol $R$, and we term the sequence the Pell cascade. Just as in the period-doubling cascade, successive terms in the Pell cascade are reached through increasing the value of the parameter $\mu$ in the logistic map of Equation (2). The first few values, and the limit $P_\infty$, are indicated in Figure 4.

The values of the parameter $\mu$ used to generate successive Pell words as stable attracting orbits in the logistic map are given in Table 2, found using the word lifting technique. In

Table 2: The sequence of parameters $\mu_C$ required to give successive superstabilized Pell words $P_n|_C$, of length $P_n$, as superstable attractive orbits in the logistic map of Equation (2). The first two values are known analytically. The others are found via word lifting, and are accurate to at least the number of digits stated. The lower and upper bound of each periodic window, $\mu_\pm$, is also stated; the period-70 window is smaller than machine precision ($10^{-16}$).

| $P_n$ | $\mu_C$ | $\mu_-$ | $\mu_+$ |
|---|---|---|---|
| 1 | 0 | 0 | 3/4 |
| 2 | 1 | 3/4 | 5/4 |
| 5 | 1.625413725123 | 1.62443 | 1.62838 |
| 12 | 1.66964217697186 | 1.66964 | 1.66965 |
| 29 | 1.67028686872861 | $\mu_C - 3 \times 10^{-10}$ | $\mu_C + 2 \times 10^{-9}$ |
| 70 | 1.67028763874509 | - | - |
| $\infty$ | $\approx 1.670288$ | | |

order to give a faster convergence of the numerical iterations, the final letter of each word has been substituted with the central point $C$, which makes the corresponding orbit superstable. The word itself can be found by infinitesimally increasing or decreasing $\mu$ so as to undo this substitution (by appeal to the periodic window theorem). The values of $\mu$ are converging on a value of $\mu_\infty = 1.6703\ldots$. This rapid convergence is a consequence of the fact that the Pell numbers $P_n$ grow faster than $2^n$, which itself follows from the additional letter in the symbol $L$ Pell substitution compared to the period-doubling substitution. The Clapeyron cascade accelerates more rapidly still. While the widths of the periodic windows are rapidly decreasing, in general the window widths are known to be system-dependent, and in any case cannot be said to simply decrease with increasing word length [33, 34].

Note that the successive Pell words do not appear contiguously, as do the successive words in the period-doubling cascade. Periodic windows in a chaotic regime are entered via the intermittency route to chaos, and exited through a period doubling cascade. All the words within one window therefore take the form of a word with the period doubling substitution rules applied to it. It follows that each Pell word longer than $RL$ appears within its own periodic window in the chaotic regime, and the sequence cannot be contiguous. As a result, although the values of $\mu$ required to generate each successive word increase monotonically, there may exist additional admissible words between each iteration which are not Pell words. It is possible to find all admissible words between two words by appeal to the periodic window theorem [34]. We omit the details here, but note that even within the chaotic regime, there are still non-Pell words interspersed between the Pell words. For example,

$$P_3 \prec RLR^2LRLC \prec P_4 \prec P_4RLR^2LRLC \prec P_4RLR^2LRC \prec P_5 \prec \ldots$$

where '$A \prec B$' indicates that word $A$ appears at a lower $\mu$ than word $B$ in the logistic map.

The inflation property of quasilattices is key in the present study. Each unit cell $R$ and $L$ corresponds to a single iteration of discrete time in the dynamical system, and so these cells are of the same length. A single inflation leads to the cells $RL$ and $RRL$. These are also perfectly good unit cells which can be used to tile the Pell quasilattice, but are now of different lengths, two and three. This already suffices to define the infinite Pell word, appearing as a stable attracting orbit in discrete-time dynamical systems, as a time quasilattice: a tiling of the time axis by unit cells of two different durations, forming an aperiodic sequence generated as a slice through a periodic tiling of two-dimensional time. Should it be desired that the durations of the cells, in addition to their frequency of appearance, also lie in the silver ratio, systematic approximations can be formed by repeated action of the inflation rules.
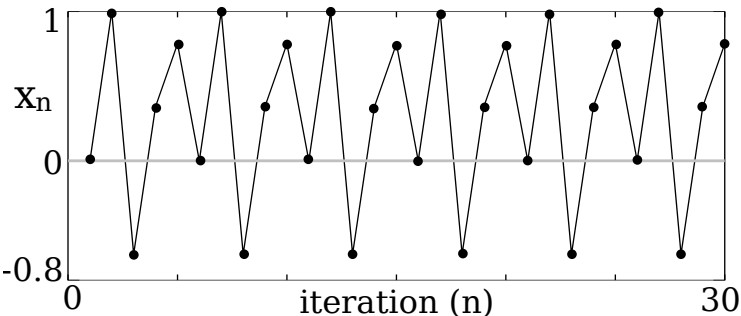

Figure 5: The sequence of points $x_n$ visited in the logistic map with $\mu = 1.625\ldots$ chosen to give the superstabilized period-5 Pell word $P_3$.

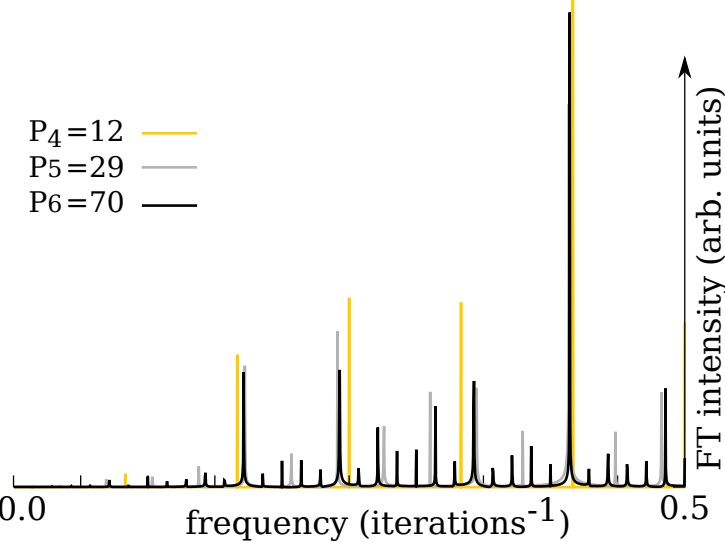

Figure 6: Temporal Fourier transforms of the sequences of points visited for superstable Pell words $P_4$, $P_5$, $P_6$ (periods 12, 29, 70). 3000 points are taken in each run. The Bragg peaks are converging on the quasilattice values, demonstrating that the successive Pell words constitute a systematic approximation scheme to the infinite word, while a dense background begins to form demonstrating the quasilattice nature of the infinite word.

In Figure 5 we show the sequence of points $x_n$ visited in the first 30 iterations of the logistic map with $\mu = 1.625\ldots$ chosen to give the superstabilized period-5 Pell word $P_3|_C$. In Figure 6 we show the temporal Fourier transform of the first 3000 iterations for Pell words $P_4|_C$, $P_5|_C$, and $P_6|_C$ of periods 12, 29, and 70, respectively. The plots demonstrate two important points. First, the largest Bragg peaks shift only slightly between successive words. They are converging on their locations in the infinite Pell word, showing that the successive finite words indeed form a systematic approximation to the quasilattice. The largest Bragg peak is converging on $\left(1 + \sqrt{2}\right)^{-1}$, as expected for the infinite Pell word which is described by two cells with lengths related through the silver ratio $1 + \sqrt{2}$. Second, the longer words begin to develop a dense background in addition to the Bragg peaks. This is expected for quasilattices, which lie between periodicity, which would show sharp Bragg peaks, and disorder, which would show a dense, uniform distribution.

### 4.4 Other Time Quasilattices

Other than the infinite Pell and Clapeyron words, no other quasilattices can possibly be grown as stable attractive sequences in dynamical systems, as the other cases fail to fulfill the generalized composition rules.

It could always be the case that the infinite words describing the other eight quasilattice classes happen to be stable. For example, the eight-letter Fibonacci word *RLRRLRLR* happens to be stable despite the Fibonacci inflation rules not fitting the composition criteria. Other examples exist, and we cannot rule out the possibility that the Fibonacci quasilattice is also stable. Without a systematic growth rule it is not clear how this could be seen. Even if the infinite word could be shown to be stable but lacking in a systematic growth mechanism, the result would not be testable in any finite-duration experiment, so would be physically uninteresting.

Nevertheless, taking again the example of the infinite Fiboncacci word, searching along its length we can find many sub-words of any desired finite length. If we find a sub-word which is admissible as a stable periodic orbit, it could still be used as a periodic approximation to the Fibonacci quasilattice. In effect, we would simply be starting the Fibonacci word at a different point in time. To construct a set containing all the possible sub-words (and more) we note that the Fibonacci inflation rules imply that neither two *L*s nor three *R*s are ever adjacent in the word. We could then consider all possible words of a given length consisting of the letters *L* and *R* and excluding words according to this observation. Actually, an infinite hierarchy of such restrictions is necessary: it is also true that neither two blocks of *RL* nor three blocks of *RLR* are ever adjacent, for instance. Such systems are considered elsewhere, for example in references [30, 55], where they are treated by various techniques of combinatorics.

Sub-words of the infinite Fibonacci word can be found which describe stable admissible orbits of most lengths. The number of distinct $n$-letter sub-words of the Fibonacci quasilattice is $n + 1$. Searching all possible sub-words up to length 500, the cumulative total number of admissible words appears to grow as a logarithmic Devil's staircase, as shown in Fig. 7. However, even in the instances of admissible sub-words, we re-iterate that without a systematic growth mechanism such as we have found for the Pell and Clapeyron quasilattices, the result is uninteresting from a physical point of view.

## 5 Pell Words in Continuous-Time Dynamical Systems

The work so far has focussed on the logistic map of Eq. (2), a discrete-time dissipative dynamical system [37]. In this section we generalize our results to a range of continuous-time dynamical systems, in order to provide physically testable examples.

We proceed as follows. In Section 5.1 we consider a continuous-time dissipative system without driving, the Rössler attractor, and show that the results obtained for the logistic map extend to this system. Lacking predefined time steps, however, the duration of each step ceases to be stable to perturbations, and the result is no longer a time quasilattice in the desired sense. In Section 5.2 we consider a continuous-time periodically-driven dissipative system, the forced Brusselator, in which the length of each period of the time quasilattice is fixed by the periodicity of the external driving. This constitutes a time quasilattice in a continuous-time dissipative system. In both cases the systems considered are specific cases used to illustrate much wider classes.

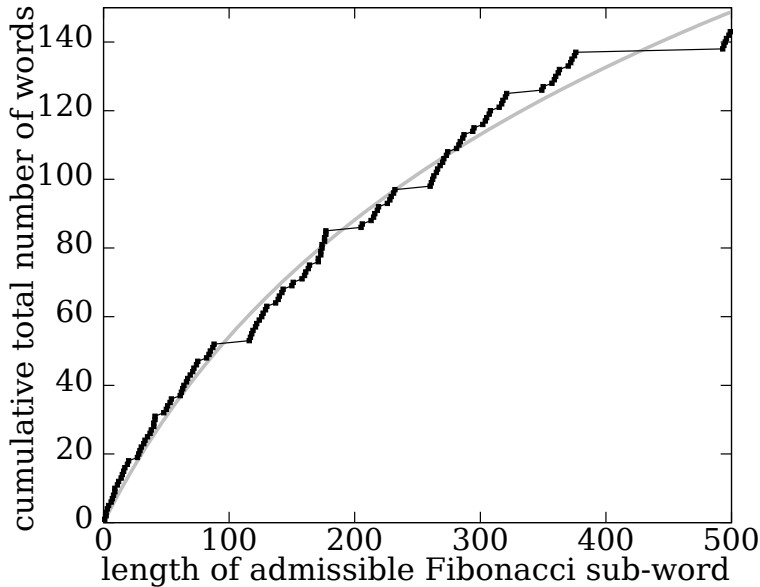

Figure 7: Black points: the cumulative total number of admissible words (in the sense of the universal sequence) found in the sub-words of the Fibonacci quasilattice up to a given length. Black lines connect the points. Silver curve: logarithmic fit to the data, $y = 93.1 \log(1 + x/126.6)$. There are $n + 1$ sub-words of the Fibonacci quasilattice of length $n$, so the cumulative total number of sub-words up to length $n$, neglecting admissibility, is $\frac{1}{2}(n+1)(n+2)$.

## 5.1 A Continuous-Time Dissipative Autonomous System: the Rössler Attractor

The Rössler system is defined by the continuous-time equations of motion:

$$\dot{x}(t) = -y - z$$
$$\dot{y}(t) = x + ay$$
$$\dot{z}(t) = b + z(x - c). \tag{7}$$

The system is dissipative. For a wide range of parameters the system features a strange attractor. The case of $a = 0.2$, $b = 0.5$, $c = 5.7$ is shown in Fig. 8. We define a Poincaré section through the attractor by finding its intersections with the half-plane $x = 0$, $y > 0$. The trajectories' intersections with the plane are shown in the figure.

We use the Poincaré section to construct a Poincaré first return map, plotting the $y$ co-ordinate at intersection $n + 1$, $y_{n+1}$, against the $y$ co-ordinate of the previous intersection $y_n$. The map is unimodal, bearing a strong resemblance to the logistic map. The first return map is shown for three values of the parameter $b$ in Fig. 9. Note that the maps are well-approximated by 1D lines, which follows from the large dissipation in the Rössler equations of motion.

Varying $b$ over the range $0 < b < 2$ we find all the qualitative features derived for the logistic map. In Figure 10 we plot an orbit diagram for the Rössler system by plotting $y_{5000}$ to $y_{8000}$ (it is assumed stable orbits have been reached after this many iterations) against $2 - b$. Chaos develops via a period-doubling cascade reached by around $2 - b = 1.3$. The universal sequence of periodic windows within the chaotic regime is again visible, with the period-6 orbit just below $2 - b = 1.4$, and the period-3 orbit at around $2 - b = 1.65$. The maxima in the first-return maps vary as a function of $b$, and we have located and plotted them on the orbit diagram in black. These constitute the superstable points in the orbits. Note that each periodic window again has a superstable point contained within it.

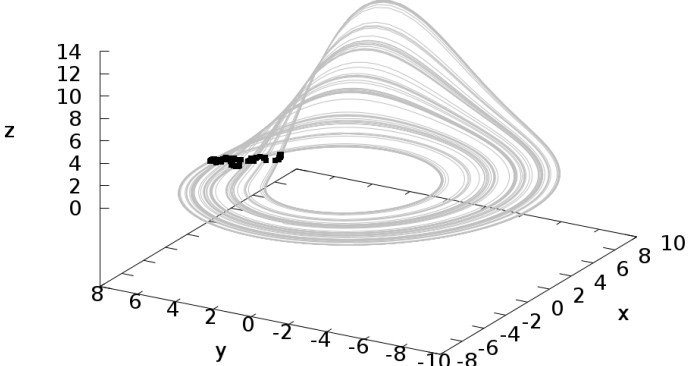

**Figure 8:** The Rössler strange attractor given by Eq. (7) for parameter values $a = 0.2$, $b = 0.5$, $c = 5.7$. The black points mark the intersections with the Poincaré section given by the half-plane $x = 0$, $y > 0$. We iterated the equations of motion, Equation (7), using a fourth-order Runge-Kutta algorithm.

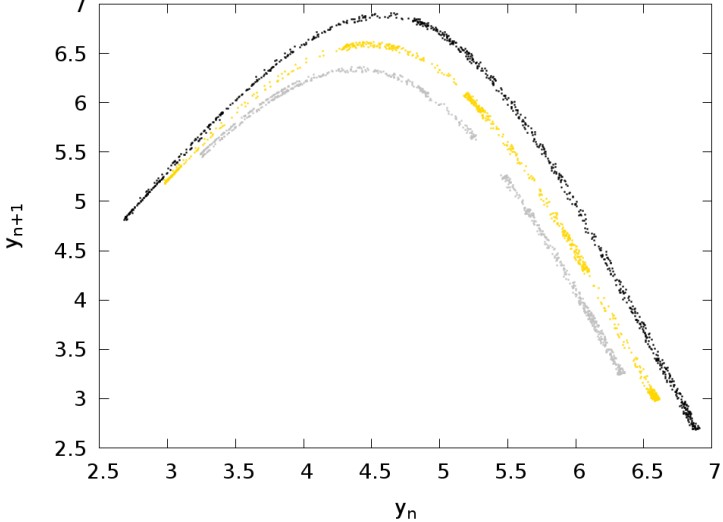

**Figure 9:** The Poincaré first-return map obtained from the Rössler equations of motion (*cf.* Fig. 8) by plotting the $y$ co-ordinate, $y_{n+1}$, of intersection $n+1$ with the Poincaré section $x = 0$, $y > 0$, against the $y$ co-ordinate of the previous intersection $y_n$. The parameters are $a = 0.2$, $c = 5.7$, $2-b = 1.4$ (silver), $2-b = 1.5$ (gold), $2-b = 1.6$ (black). Note that the maxima shift slightly as a function of the varying parameter $b$.

As an example of a Pell word in the Rössler system, the period-5 window can be seen around $2-b \approx 1.5$. Zooming in, we found the window to be located around $2-b \approx 1.492$. Setting this parameter value, the system rapidly converges to a period-5 orbit, which has intersections with the first return map described by the word $RLRRC$. A slight decrease of $2-b$ then stabilizes the Pell word $P_5 = RLRRL$. The $2-b = 1.492$ first-return map (after allowing transients to die down) is shown superposed on the close-by $2-b = 1.5$ chaotic map in Figure 11, along with

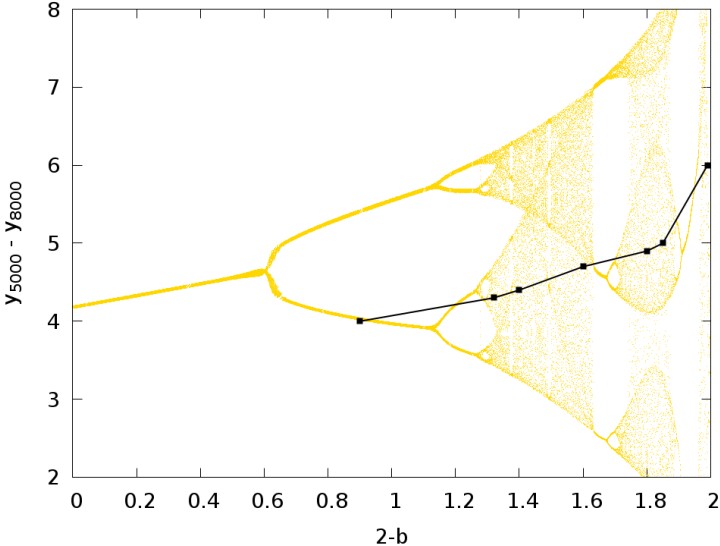

Figure 10: Orbit diagram for the Rössler system found by plotting intersections $y_{5000}$ to $y_{8000}$ of the Poincaré section for a range of parameters $b$. It is assumed stable orbits have been converged to for parameters at which they exist. The maxima of the first-return maps have been used to identify the superstable line shown in black (points indicate identified maxima). All qualitative features of the logistic map appear, including a period-doubling cascade into chaos, and a series of periodic windows following the universal sequence which have opened around superstable orbits.

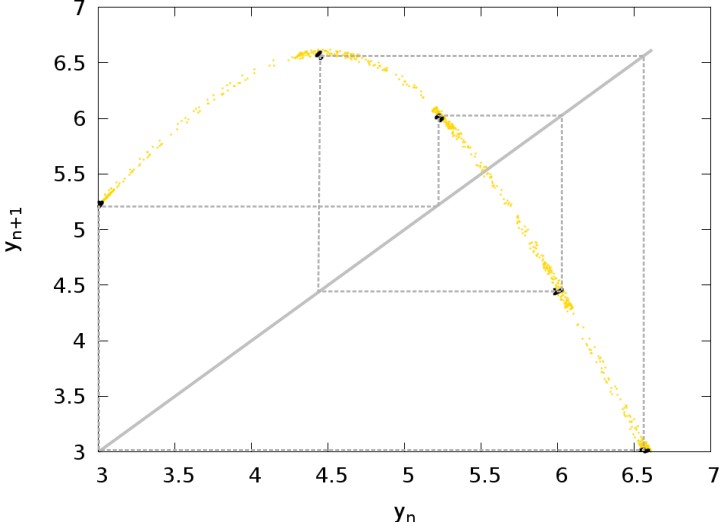

Figure 11: In gold is a reproduction of the $2 - b = 1.5$ first-return map from Fig. 9. In black is the first-return map for $2 - b = 1.492$. The slight change in parameter takes us into a period-5 window stabilized by the superstable orbit described by the superstabilized Pell word $P_5|_C = RLRRC$ (the first 5000 steps have been omitted so that the periodic orbit is reached). The cobweb has been added to emphasize how the results for discrete-time maps have carried to this continuous-time system.

the cobweb which generates it in the discrete-time 1D map. Figure 12 shows the trajectory itself.

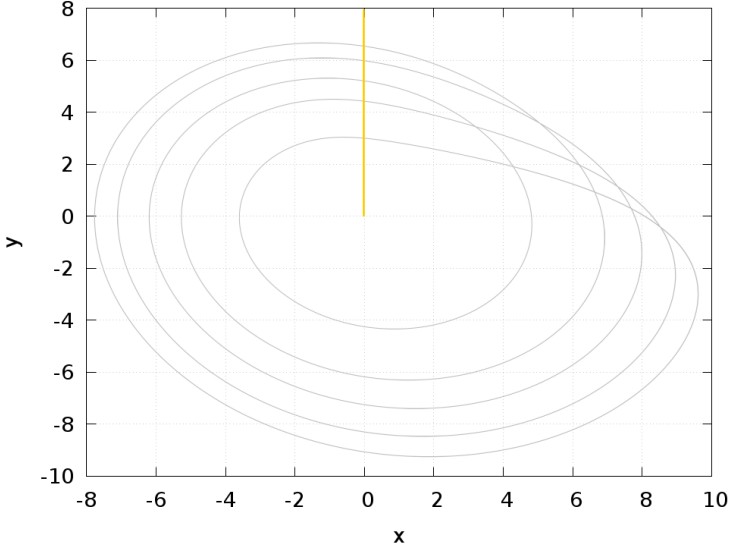

Figure 12: The projection into the $xy$ plane of the continuous-time orbit generated by the superstabilized Pell word $P_5|_C = RLRRC$, found by setting $a = 0.2$, $2 - b = 1.492$, $c = 5.7$ in the Rössler equations and iterating for 5000 steps before plotting so as to allow the orbit to stabilize. In gold is the Poincaré section.

The results match nicely with those predicted by the logistic map. From the existence of the universal sequence in the Rössler system it follows that the Pell words, including the infinite Pell word, exist as stable attracting orbits in the Poincaré first-return map. A qualitatively similar, physically-implemented system is the Belousov-Zhabotinsky autocatalytic chemical reaction [56–58]. This system, too, is known to feature the universal sequence, and therefore also features the Pell words as admissible trajectories.

However, having moved to continuous time, neither system can reasonably be described as a time quasilattice. Although the sequence of $L$s and $R$s visited by the trajectory remains fixed, the 'time of flight' between successive intersections of the Poincaré section is neither fixed, nor stable to perturbation. In order to find such stability, we consider a periodically driven dissipative system in the next section.

## 5.2 A Continuous-Time Dissipative Driven System: the Forced Brusselator

The forced Brusselator (portmanteau of 'Brussels' and 'oscillator') is described by the equations

$$
\begin{aligned}
\dot{x}(t) &= A - (B + 1)x + x^2 y + \alpha \cos(\omega t) \\
\dot{y}(t) &= Bx - x^2 y.
\end{aligned}
\tag{8}
$$

The undriven model with $\alpha = 0$ has been used to model certain autocatalytic chemical reactions, and again bears similarity to the Belousov-Zhabotinsky reaction [57–59]. Featuring an external driving frequency $\omega$ the equations are non-autonomous, although they can be rewritten as an autonomous set of four equations by defining

$$
\begin{aligned}
u(t) &= \cos(\omega t) \\
\dot{u}(t) &= -\omega z \\
\dot{z}(t) &= \omega u
\end{aligned}
$$

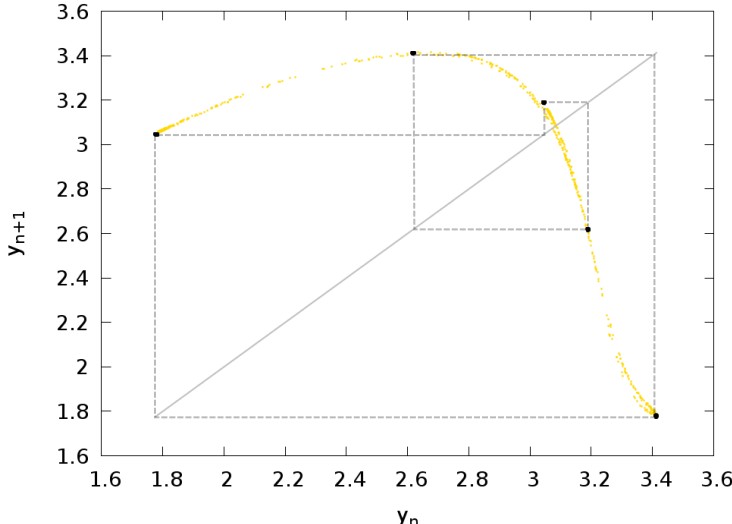

Figure 13: The Poincaré first-return map obtained from Poincaré section $u = 0$ of the forced Brusselator of Equation (8) with $A = 0.38$, $B = 1.2$, $\alpha = 0.05$. In gold are the points obtained at driving frequency $\omega = 0.72$, where the response is chaotic. In black are the points obtained at driving $\omega = 0.725$ which gives a stable period-5 orbit described by the Pell word *RLRRL*. The equations were iterated with a fourth-order Runge-Kutta algorithm iterated $10^5$ steps, discarding the first 5000 steps to allow transients to decay.

which can then be iterated using a Runge-Kutta algorithm. The natural Poincaré section to take is now any constant $u$ plane, corresponding to a periodic sampling of the system. Note that admission to the section is dependent upon the trajectory crossing with the correct orientation.

The phase space of the equations is quite complex. While it does feature period-doubling cascades as functions of various parameters, we were unable to locate any leading directly into universal sequences as in the previous examples. Nevertheless, the existence of the universal sequence words within the system has previously been confirmed [34, 59].

Taking the parameter values $A = 0.38$, $B = 1.2$, $\alpha = 0.05$, we find that a chaotic regime at frequency $\omega = 0.72$ develops into a stable period-5 window at $\omega = 0.725$. In Figure 13 we show the Poincaré first-return map obtained from the Poincaré section $u = 0$ crossed in a positive sense, for both the chaotic response (gold) and period-5 response (black). As before, we see that the trajectory is described by the Pell word *RLRRL* (this frequency is slightly away from the superstable point *RLRRC*, so the word itself appears). In Figure 14 we show the period-5 trajectory and the Poincaré section.

The forced Brusselator system has the advantage over the Rössler system that the time taken between intersections of the Poincaré section is fixed to be a multiple of the driving frequency. The cell lengths $R$ and $L$ are therefore identically equal to one period of the driving, and are stable to perturbations either of other system parameters or to random kicks to the trajectory. Subsequent inflations such as $RL$, $RRL$ constitute two different cell lengths which can be used to tile longer Pell words when located in the system, including (in principle) the Pell quasilattice.

It should be noted that the forced Brusselator does not fall into the same universality class as the Rössler system or the logistic map, as it features a cubic nonlinearity. Fortunately, this is not necessary for the words of the universal sequence to appear as stable orbits on the associated Poincaré map. In fact, the universal sequence has previously been identified in the forced Brusselator system [34]. This is important: if we know that the universal sequence appears,

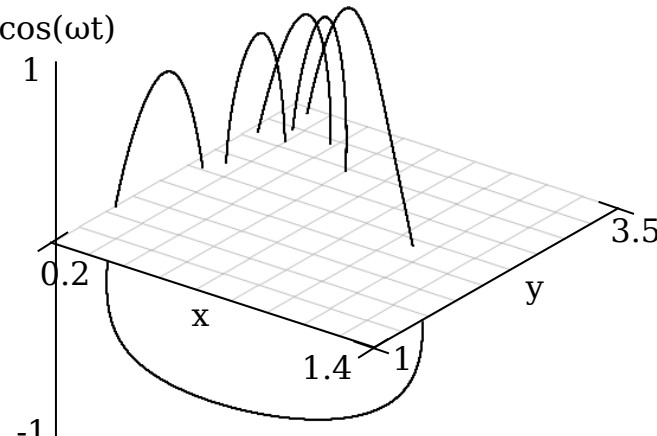

Figure 14: The trajectory followed by the forced Brusselator equations of motion in the period-5 window considered in Figure 13. The Poincaré section $u = \cos(\omega t)$ is shown (only trajectories passing from negative to positive $u$ are included).

even in a re-ordered form, it means that all Pell and Clapeyron words appear. This means we can make periodic approximations to the Pell and Clapeyron quasilattices of arbitrary duration, by refining our experimental precision. If the universal sequence does not appear, it may be that individual words can be found regardless, but the process may fail at or above a given duration. Furthermore, without some reason to believe that all the words appear in principle, we are merely investigating periodic orbits of the first-return maps, as opposed to using them as periodic approximations to time quasilattices.

The forced Brusselator is stabilized to perturbations in part by its dissipation (the equations of motion can again not be rewritten into a Hamiltonian form). Despite lacking a conserved quantity which can be thought of as energy, it is still possible to define a temperature-like external noise in dissipative systems, at least away from the superstable points. Robustness of the trajectories to finite temperature necessarily requires the interactions of a macroscopic number of degrees of freedom, not present here [21]. The next section addresses this issue.

## 6 Time Quasicrystals

In the previous sections we established the concept of time quasilattices: the mathematical structure of quasilattices, in the time direction. We found them as stable and structurally stable trajectories in dissipative dynamical systems. Until now we have not been concerned with the physical origin of the stabilising non-linearity, owing to the universality of chaotic dynamics [32]. After the present paper appeared online, a number of experimental proposals for realizing discrete time crystals in driven dissipative quantum many-body systems were proposed [19, 20]. In this section we identify signatures of time quasilattices in these systems; as the structures are additionally rigid in the same sense as time crystals, we identify these responses as new states of matter, 'time quasicrystals'. We begin by providing precise definitions of these phrases before identifying signatures of the states.

Discrete time crystals feature a period-doubled response to a periodic driving [7–10]. There should also be a sense of *rigidity*, in order to bring them in line with our intuition regarding spatial crystals [21]. The three cases of interest in the present context are as follows.

- A *time quasilattice* returns an aperiodic response to a periodic driving, featuring two unit cells of different durations, where each cell appears with precisely two spacings and the ratio of cell populations tends to a Pistot-Vijayaraghavan number as the number of cells tends to infinity. Both the durations of the cells and their sequence are stable and structurally stable to perturbations, so the order persists indefinitely.

- A *discrete time crystal* occurs when (i) the discrete time translation symmetry of a periodic driving is spontaneously broken by a lower-period response, which is (ii) made both stable and structurally stable to perturbations and finite temperature by (iii) the local interactions of many degrees of freedom, and which (iv) persists indefinitely. There should also be (v) a sense in which it can be understood to be a ground state.

- A *discrete time quasicrystal* occurs when the discrete time translation symmetry of a periodic driving is spontaneously broken by a time quasilattice response, which is made both stable and structurally stable to perturbations and finite temperature by the local interactions of many degrees of freedom, and which persists indefinitely. There should also be a sense in which it can be understood to be a ground state.

Note that this definition of time crystals does not necessarily include quantum mechanical effects; classical discrete time crystals have been proposed, and the original proposal for classical time crystals, which break continuous time translation symmetry, was not ruled out by the no-go theorems applied to the quantum case [2, 6, 21].

Some leeway is built into requirement (v), since the concept of a true ground state requires energy to be conserved, which is not the case in any of the known examples of discrete time crystals. In refs. [7–10] the periodic driving leads to a pseudo-energy being conserved modulo $2\pi$, and it is in this sense requirement (v) is fulfilled. Reference [21] uses the phrase *rigid subharmonic entrainment* for dissipative systems fulfilling the other criteria, reserving the phrase *classical discrete time crystals* for the case in which the classical many-body system remains rigid when coupled to a finite-temperature bath (although, since inherently out-of-equilibrium, the concept of a ground state is again avoided). Other references refer to these states as *dissipative discrete time crystals* [19] or equivalent phrases [20, 60]. This is the convention we adopt here.

The advantage of explicitly allowing dissipation is that states beyond period doubling can be stabilized. In reference [19] a protocol is outlined to identify dissipative discrete time crystals in quantum many-body cavity/circuit QED setups governed by the Dicke model. Numerical simulations of the classical limit show several signatures the authors identify in a simplified discrete-time nonlinear model featuring a period-doubling cascade into chaos. The authors further argue that these signatures are also present in the quantum many-body limit which would be realized by the experiments they propose. The experimental identification of a period-doubling cascade is a sufficient condition for all of the Pell and Clapeyron words to appear, and would prove that these systems feature time quasilattices. Since the stability derives from the interactions of many degrees of freedom, and the quantum many-body system features spontaneous symmetry breaking into this state, these setups would then feature true (dissipative, discrete) *time quasicrystals*.

In reference [20] dissipative discrete time crystals are identified in a numerical model of a driven open quantum system (bosonic atoms in a double-well potential). There is a clear period doubling cascade into chaos in the model's classical limit – again, a sufficient condition for the presence of time quasilattices, and therefore in this scenario (dissipative, discrete) time

quasicrystals. The authors also identify the continuation of the classical period-doubled state to the quantum regime via two-time correlation functions, opening the possibility of identifying *quantum* dissipative discrete time quasicrystals in this system.

A proposal for a non-dissipative discrete time crystal based on a kicked Lipkin-Gleshkov-Glick model is provided in [60]. This model features a Hamiltonian system of spins with a periodic driving, and the authors identify candidate experimental implementations in Bose Einstein condensates and trapped ion systems. Rigid responses of various periods are identified within a classically chaotic regime, although no period-doubling cascade is immediately obvious. Strictly, the proposals in both references [19] and [60] violate requirement (iii) above, since the many-body interactions stabilizing the discrete time crystal phases are infinite-range rather than local. Local couplings ensure that the concept of dimensionality is well-defined in abstract mathematical models: depending on the topology of local connections, a model of many-body interactions could correspond to a range of physical dimensions. In the present context of physical interacting particles, however, this requirement seems unnecessarily limiting (the systems are all three dimensional).

In all these cases, the procedure for identifying the time quasicrystals in the classical regime would be to identify the sequence of periodic approximations (the finite-length Pell or Clapeyron words) as an externally-tunable field is varied. The external field depends on the individual systems [19, 20, 60]. Several Pell and Clapeyron words can already be seen without further analysis in the classical limit of reference [20]. The time quasicrystals themselves are indistinguishable from any of their periodic approximants featuring a period longer than the observation time. Nevertheless, their existences and stabilities are guaranteed by the periodic window theorem [59].

In experimental searches for time quasicrystals, the only things which can be measured are periodic approximations. This restriction is made necessary by the finite duration of the experiment. Each periodic response is simply a dissipative discrete time crystal, and so the techniques developed in the references already suffice to identify them in both the classical and quantum regimes. The only extension necessary experimentally would be to identify that, as a function of the tunable system parameters, a sequence of time crystals is found with periods increasing as either the Pell or Clapeyron words.

# 7 Conclusions

In this paper we have demonstrated the existence of time quasilattices in dissipative dynamical systems. These are aperiodic tilings of the time axis with two different unit cells, which can be constructed as slices through two orthogonal time directions. We demonstrated that time quasilattices can appear as stable, attracting orbits in any dissipative nonlinear dynamical system which features the universal sequence, or any re-ordering thereof [34, 39]. This is a wide universality class, encompassing physical applications in the study of animal populations [38], chemical reactions [57,58], hydrodynamics [61], and electronics [62,63], to name a few. We demonstrated that these time quasilattices can be 'grown' by repeated application of their inflation rules, meaning that systematic finite periodic approximations can be found in any system in which they exist, giving an experimentally-testable method of searching for them.

Of the ten equivalence classes of physically-relevant one-dimensional quasilattices recently identified by Boyle and Steinhardt [28], we find that precisely two are able to form time quasilattices. These are class 2a, the infinite Pell word $P_\infty$ related to the silver ratio $1 + \sqrt{2}$, and class 3a, the infinite Clapeyron word $C_\infty$ related to $2 + \sqrt{3}$. The relevance of these cases is that they generalize to higher-dimensional quasicrystals which can be grown in the lab [24,

25]. Class 2 generalizes to the Ammann-Beenker tiling, a two-dimensional tiling with an 8-fold rotationally-symmetric diffraction pattern, and class 3 generalizes to the cases of two-dimensional tilings with 12-fold rotationally-symmetric diffraction patterns [23].

It is necessary to consider the time quasilattices' place in the set of periodic space-time orders discussed in the Introduction. Choreographic crystals constitute an extension of the concept of space group symmetry to encompass the time direction [13]. The interesting cases contain multiple moving elements, whereas the time quasilattices discussed here focus on the trajectory of a single particle. Interestingly, chaotic systems can demonstrate synchronization when coupled, while maintaining their unpredictability [32, 48, 64]. This synchronization can take the form of a fixed delay between points on the particles' trajectories; it has even been demonstrated to persist to the quantum regime of systems with a chaotic classical limit [65–67]. Since both periodicity and chaos can synchronize it would seem likely that time quasilattices, lying between the two, can do so as well. This would suggest a multi-particle implementation of dissipative time quasilattices which could feature an enhanced choreographic order.

Discrete time crystals spontaneously break the discrete time translation symmetry of a periodic driving force with a response of twice the period [7–10]. They are also required to be stabilized to finite temperature through the interactions of many degrees of freedom, fitting our intuition for what constitutes a 'state of matter'. There should also be some sense in which they can be considered the ground state of a system, as with crystals in space. Several experimental proposals have recently appeared for discrete time crystals in dissipative systems [19, 20]. Numerical simulations show clear examples of period-doubling cascades into chaos in the systems' classical limits. This is a sufficient condition for realizing time quasilattices. If identified experimentally these systems would therefore feature (dissipative, discrete) *time quasicrystals*: the discrete time translation symmetry of a periodic driving is spontaneously broken to the symmetry of a time quasilattice, which is stabilized against perturbations via the local interactions of a quantum many-body state. They can be experimentally identified by their sequences of periodic approximations (finite Pell and Clapeyron words).

It is interesting to consider what steps would need to be taken to identify time quasilattices in a non-dissipative (Hamiltonian) context. A Hamiltonian system would need to be located with a sufficiently one-dimensional Poincaré first-return map, which demonstrates the universal sequence of admissible words. The system would either need to be constructed from a macroscopic number of degrees of freedom, or a macroscopic number of such systems would need to be coupled, such that the interactions between the degrees of freedom stabilize the order to finite temperature. We note that dissipation is not the key criterion for the existence of an effective one-dimensional Poincaré first-return map: hyperbolicity is [30,55]. This requires that the flow have at least one positive, one negative, and one zero Lyapunov exponent, with the associated eigenvectors giving the directions of the unstable, stable, and marginal manifolds of the flow [43,44]. The proof of such a property has not even been rigorously established for the Rössler map, but numerical evidence for it exists in a wide range of systems, including Hamiltonian [33, 48, 55]. The universal sequence, as its name suggests, is ubiquitous. Together, these facts suggest the identification of time quasilattices in Hamiltonian systems may be an achievable goal.

From a theoretical perspective, perhaps the most exciting application of time quasicrystals could be as a test of theories concerning multiple dimensions of time. Higher dimensions of space are routinely discussed in the theoretical physics literature, and quasicrystals have been proposed as implementations in several recent papers [68–72]. Higher time dimensions are occasionally considered, but have received relatively less attention [73–75]. It is our hope that the present suggestion of the possibility of testing such theories will lead to wider discussion of these ideas.

# Acknowledgements

The author wishes to thank M. V. Berry, J. P. Keating, M. A. Porter, K. Sacha, J. van Wezel, N. Y. Yao, and M. Zaletel for many helpful discussions, and L. Boyle for both helpful discussions and a critical reading of the manuscript.

**Funding information** The author acknowledges support from a Lindemann Trust Fellowship of the English-Speaking Union and the Astor Junior Research Fellowship of New College, Oxford.

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
