# Peer review of "Time Quasilattices in Dissipative Dynamical Systems"

_SciPost Physics, doi:SciPost Phys. 5, 001 (2018)_

## Round 2 · Referee Report · Anonymous · 2018-2-20

Strengths
1. The paper adresses a timely subject.
2. The presentation is very clear, I actually enjoyed reading it.
3. It gives a thorough discussion on time quasicrystals
Weaknesses
1. In terms of experimental implications, the discussion is too concise.
2. It is not clear to me how these findings would appear for the quantum version of time quasicrystals. This, I feel, is an important issue as existing experiments for time crystals stem from quantum systems.
Report
The author discusses time quasicrystals in dynamical dissipative systems. This is a very timely subject, and since the original proposal for quantum and classical time crystals, there was an explosion of interest in this direction, followed by the experimental realization of such systems last year. Similarly to how crystals and quasicrystals exist in solid state physics, the author generalizes the concept of time crystals to time quasicrystals as stable trajectories in dissipative dynamical systems. Using the generalized composition rules, two admissible quasicrystals coined as Pell and Clapeyron quasicrystals are identified. In terms of physical models, discrete-time maps as well as periodically-driven continuous-time dynamical systems are analyzed.
Requested changes
1. Although a short listing of experiments is given in terms of where time quasicrystals could be looked for, a bit more detailed discussion about more specific experiments and signatures would be helpful for a general reader.
2. The author identifies two classes of quasicrystals (Pell and Clapeyron), characterized by different ratios of the cell length as 1+sqrt(2) and 2+sqrt(3).
Just by looking at these numbers, is it possible to have a quasicrystal with ratio for word length 4+sqrt(5) or something similar?
3. A discussion on the quantum version of time quasicrystals and their respective signatures would also be interesting.
Krzysztof Sacha on 2018-03-03 [id 221]
Time crystals are recently invented phenomena which originally have been associated with spontaneous self-organization or self-re-organization of quantum many body systems in a periodic way in time in full analogy to spontaneous formation of periodic crystalline structures in space in condensed matter physics. The original time crystals' idea proposed by Frank Wilczek triggered a new field of research. I would differentiate two branches of this field: investigation of systems which can start spontaneously periodic motion and modeling condensed matter physics in the time domain. The work by Felix Flicker opens up a new direction, in my opinion. That is, the research of non-trivial crystalline structures in time in classical dynamical systems. The existence of periodic orbits in time evolution of a classical dynamical system is not very surprising. However, the fact that it is possible to find trajectories of classical systems that reveal “quasi-crystal tilling of time” is something new and interesting. The strategy performed by Flicker is not just to find a quasi-periodic behavior of a system. He presents a systematic approach which allows one to find parameters of the systems which ensure that any finite Pell or Clapeyron quasi-crystals can be reproduced by stable periodic trajectories of a classical dissipative system.
The considered systems are classical and dissipative and it is not easy to quantize them as well as it is not easy to realize similar behaviour in time evolution of a quantum particle. Actually, I do not see a possibility for a direct analogy of the presented classical behaviour in the quantum word. I think that in the quantum case, time quasi-crystals must be defined in a different way but maybe I am wrong. Anyway the paper will definitely become an inspiration for other scientists as it has been for me.

---

## Round 2 · Referee Report · Anonymous · 2018-3-19

Strengths
1) Good introduction to quasi-crystals and symbolic dynamics
2) Creative and interesting idea to combine the two concepts
Weaknesses
1) Connection to time-crystals not sufficiently clarified (see report)
Report
Generally, dissipative driven systems can have periodic orbits as stable attractors. Such periodic orbits can be stable (with respect to small parameter changes in the dynamical system) and attractive. If these attractors are course-grained (e.g., the point after each period is labelled according to whether it is "left" or "right" of the origin of a one-dimensional coordinate) one obtains a sequence of labels (a "word") that characterises the dynamics. In the manuscript it is shown that there exist two sequences of words of increasing length ("Pell words" and "Clapeyron words"), each corresponding to stable and attractive orbits, which converge to one-dimensional quasicrystals when "L" and "R" are identified with the short and long cells in the quasicrystal. In the manuscript these sequences are termed time-quasicystals.
However, in the present form of the manuscript I do not see a strong connection to "time-crystals":
In the recent literature, a time-crystal refers to the breaking of (discrete) time-translational invariance in a periodically driven quantum system. In the present manuscript, the notion is extended to classical dissipative systems. Before going to time-quasicrystals, a clear definition should therefore be given what is a normal time-crystal in the present setting. Should any periodic orbit in a driven system be considered a time-crystal if it is an attractive trajectory which is stable against perturbations of the parameters? With this, "time-crystals" would to be a common phenomena in dynamical systems which does not really need a new terminology. Is there an additional rigidity that characterises this "phase of matter"?
The aperiodic orbits of quasi-crystalline order which are discovered in the manuscript seem to satisfy even less the requirements for being a "stable phase", as they are not even stable against parameter changes of the dynamical system. While it is shown in the manuscript that any periodic approximation of the discussed time-sequences is stable throughout an open set of parameter space, in the discussed examples this stability range decreases strongly with length (e.g., in the logistic map the period-12 approximation to the Pell quasicrystal is stable only in a parameter range of $10^{-5}$). Even though, as the author says, there is no generic proof that the stability range would decrease with the length of the approximation, no example for a "time-quasicrystal" which is stable in a finite parameter regime is presented. Instead, the infinite time-quasicrystal seems to be embedded in a chaotic region of the dynamics which would be destroyed by infinitesimal variations of the parameters of the dynamical system.
I believe that a notion of rigidity should be added to the definition of time-crystals in dynamical systems, otherwise the term should probably be avoided (also in the title). In the paper it would still be useful to add a proper discussion of the differences between time crystals and periodic orbits in dynamical systems, which would make the paper still appealing for researchers working on time quasicrystals.
Furthermore, I have one minor technical question: The manuscript states that *only two* out of ten possible quasicrystals an be realised as course-grained time-series in the logistic map. Following the definitions given in section 3.4, I understand that if inflation rules that lead to a quasicrystal respect the the generalized composition criteria, we have managed to find a sequence of stable and attractive orbits of increasing length which finally gives a quasi-crystal order. While this can be used for a constructive proof for the existence of the two classes (Pell and Clapeyron), I do not yet see how it excludes other classes (probably because of my limited knowledge on symbolic dynamics). The generalized composition rules are only stated as a *sufficient* criterion to produce a sequence of maximal words, which does not exclude that inflation rules which do not satisfy them can produce an infinite sequence of maximal words (as I understand, it is not needed for every word in the series to be maximal in order to produce a series which approaches the quasi-crystal?).
In any case, the paper establishes a connection between two different concepts (quasi-crystalline order and classical dissipative driven systems). I found this an interesting observation, and it was a nice to think about it. The discussion is also presented in a detailed and pedagogic manner, which makes the manuscript accessible to readers who are not from the field of dynamical systems. Hence I believe the paper should be published if the discussion on time crystals is adapted.
Requested changes
see report

---

## Round 3 · Author Response

I would like to thank both referees for their very positive reports, and Prof. Sacha for taking the time to comment on the work. I would also like to thank the editor-in-charge for managing this submission. The suggestions by both referees were very helpful, and I think the paper is much improved from their implementation. I will discuss the broad outline of the changes I have made, before giving specific responses to each reviewer, followed by a list of specific alterations.
outline of alterations
Both referees requested that I clarify the relationship between the infinite Pell and Clapeyron words, and time crystals. To this end, I have made two substantial changes. First, I have renamed the infinite Pell and Clapeyron words 'time quasilattices', rather than time quasicrystals. I feel the new phrase is better in three regards:
i) it is intended to mirror the relationship between a lattice and a crystal: the former is a mathematical construction, whereas the latter is a physical structure, with the symmetries of the former, which has the additional connotation of rigidity at finite temperature facilitated by the interactions of many degrees of freedom.
ii) This nomenclature happens to agree with a common (albeit non-universal) usage in the quasicrystal community, in which 1D hosts only quasilattices, whereas 2D, 3D, and 4D can feature quasilattices or quasicrystals. This then allows a precise definition of 'quasicrystal' as a physical system which has a diffraction pattern with a rotational symmetry forbidden by the crystallographic restriction theorem. Since rotations cannot be defined in 1D this definition disallows quasicrystals in this case.
iii) The definition of the infinite Pell and Clapeyron words as 'time quasilattices' leaves open the possibility of defining 'time quasicrystals' as systems with the same symmetry, but which are stabilised by the local interactions of many degrees of freedom, which are reached through spontaneous symmetry breaking, and which can be considered the ground state of a system. This brings the definition in line with that of time crystals (although runs counter to point (ii), which I make clear in the text).
The second major change is to include a new section, 'Time Quasicrystals', in which I give an explicit definition time quasicrystals and identify several examples in recent experimental proposals. The definitions are as follows.
-
A time quasilattice returns an aperiodic response to a periodic driving, featuring two unit cells of different durations, where each cell appears with precisely two spacings and the ratio of cell populations tends to a Pistot-Vijayaraghavan number as the number of cells tends to infinity. Both the durations of the cells and their sequence are stable and structurally stable to perturbations, so the order persists indefinitely.
-
A discrete time crystal occurs when the discrete time translation symmetry of a periodic driving is spontaneously broken by a lower-period response, which is made both stable and structurally stable to perturbations and finite temperature by the local interactions of many degrees of freedom, and which persists indefinitely. There should also be a sense in which it can be understood to be a ground state.
-
A discrete time quasicrystal occurs when the discrete time translation symmetry of a periodic driving is spontaneously broken by a time quasilattice response, which is made both stable and structurally stable to perturbations and finite temperature by the local interactions of many degrees of freedom, and which persists indefinitely. There should also be a sense in which it can be understood to be a ground state.
With these changes I have sought to address all points raised by the referees. I will now respond to the referees' specific comments in turn.
response to referee 1
I would like to thank the referee again for such a positive report.
I have added a new section, 'Time Quasicrystals', in which I address the weaknesses listed by the referee. I provide a precise definition of discrete time crystals, reproduced above, and outline recent experimental proposals to implement discrete time crystals in dissipative systems. Three of the references I have added [Russomanno PRB 95 214307 (2017), Gong et al. PRL 120 040404 (2018), Wang et al. PRE 97 020202 (2018)] consider quantum many-body systems which are chaotic in the classical limit (the first system is Hamiltonian, the second two dissipative). I demonstrate that the dissipative systems can realize time quasilattices in their classical limits. As these states are made stable and structurally stable to perturbations and finite temperature by the interactions of many degrees of freedom, they additionally fulfill the criteria for (classical, discrete, dissipative) time quasicrystals. The references indicate how the broken symmetry states they identify in the classical limit can be tracked into the quantum regime, for example using two-time correlations, meaning that they also feature (discrete, dissipative) quantum time quasicrystals.
I will now address the specific requested changes.
-
I have provided a significantly increased discussion of experimental systems in which discrete dissipative time quasicrystals can appear, namely those implementations outlined in references [Gong et al. PRL '18, Wang et al. PRE '18]. These references, as well as [Russomanno PRB '17, arXiv:1801.02628], focus on numerical studies of mathematical models, but clearly state how to implement the models physically, with precise experimental protocols. For the sake of brevity I leave these discussions of implementation to the references themselves, as I did not feel it added to the present discussion to reproduce the details. At the end of the new section I include a discussion of the signatures to seek within these setups: a series of discrete dissipative time crystals with lengths given by the Pell or Clapeyron numbers, which can be tuned between by varying the relevant external parameters.
-
The referee's suggestion that there might be a ladder of 1D quasilattice states governed by increasing quadratic irrational numbers is correct. The ladder is the infinite sequence of quadratic irrational PV numbers, and each rung corresponds to a different 1D quasilattice in the most general sense. However, Boyle and Steinhardt [arXiv 1608.08220] outline more restrictive criteria for the subset of 1D quasilattices which should be considered physically relevant, finding ten. It is this set on which I focus in the paper. As a result, by these criteria no further time quasilattices can exist.
It would indeed be of interest from a mathematical viewpoint to seek dynamical trajectories corresponding to PV numbers outside this set (note that 4+sqrt(5) is not PV, as its Galois conjugate 4-sqrt(5) does not have modulus between 0 and 1). One difficulty is that, while the PV number dictates the growth of populations of each letter upon inflation, it does not specify the order of the letters or the inflation rule itself, both of which would be needed in order to apply the present analysis. This issue is not insurmountable, and I think it would be an interesting topic for further, more mathematically-oriented, work.
-
The suggestion that I discuss the quantum case is also a very good one. I do so in the new 'Time Quasicrystals' section.
I give a precise definition of discrete time crystals, reproduced above, which follows [arXiv:1801.02628] in allowing for both quantum and classical examples. References [Russomanno PRB '17, Gong et al. PRL '18, Wang et al. PRE '18] feature discrete time crystals in quantum many-body systems, but also show how these states arise from chaotic classical limits. They include a discussion of the quantum observables which most clearly indicate the continuation of the classical phenomena into the quantum regime. In the classical limits they either identify period-doubling cascades [Wang et al. PRE '18], show clear signatures of such [Gong et al. PRL '18], or show a wide range of responses of different periods [Russomanno PRB '17]. The period doubling cascade is a sufficient condition for the time quasilattices to appear, and since the stabilising nonlinearity is originating from the interactions of quantum many body states, these in fact constitute quantum discrete dissipative time quasicrystals by the definitions above.
response to referee 2
I would again like to thank the second referee for a positive and balanced report.
The referee's suggestion that 'time quasicrystal' should be reserved for systems with rigidity is fair. I have rephrased so as to draw a distinction between what I now term a 'time quasilattice', and a rigid time quasicrystal. The definitions, which I reproduce above, are chosen to be maximally strict, covering all requisites established in previous work. I then explain which conditions must be relaxed in different scenarios - for example, no discrete time crystal is a true ground state of a system, although this was a condition of the original definition of true time crystals.
The majority of the paper now concerns 'time quasilattices' with these definitions. When the stability additionally derives from the interaction of many degrees of freedom, this makes the system stable not only to perturbations, but to finite temperature, which is a stronger statement. This then is the rigidity required to consider the system a 'state of matter', and, along with the other specifics of the definition above, we term this stronger result a discrete dissipative time quasicrystal. Since the periodic responses identified in the cited papers are termed dissipative discrete time crystals (or equivalent phrases) by the authors, it is reasonable that the time quasilattice responses they contain (but which were not identified by the authors) should be called dissipative discrete time quasicrystals.
As discussion of dissipative discrete time quasicrystals is limited to the new section, I agree with the referee that it is reasonable to change the title of the paper to instead use the phrase 'quasilattice'.
With regard to specific points raised by the referee:
-
"In the paper it would still be useful to add a proper discussion of the differences between time crystals and periodic orbits in dynamical systems"
I now address this with the additional section 'Time Quasicrystals' (see also the discussion above and in the response to referee 1).
-
"no example for a "time-quasicrystal" which is stable in a finite parameter regime is presented. Instead, the infinite time-quasicrystal seems to be embedded in a chaotic region of the dynamics which would be destroyed by infinitesimal variations of the parameters of the dynamical system."
I agree this is an important point. I am not sure how one could state an example of a time quasilattice, since, as with the aperiodic word terminating the period-doubling cascade, the specific parameter values realising the aperiodic words are system-dependent, and are likely irrational and impossible to state precisely. Upper and lower bounds of arbitrary finite precision can be given.
I disagree that the quasilattices would be destroyed by infinitesimal parameter changes. As the referee notes, in any periodic case, the periodic window theorem guarantees that the window surrounding the word is of finite measure, so would survive an infinitesimal perturbation. Arbitrary aperiodic words are not covered by the periodic window theorem, since unstable ergodic trajectories spell aperiodic words which form counterexamples. However, I am confident that maximal aperiodic words are covered by the periodic window theorem. To see this, note that the generalised composition rule proves the existence of the superstable aperiodic word, with its final letter substituted with a C. Replacing the C with an L or R again returns a stable but not superstable word, and continuity implies that these form a finite-measure window around the superstable word.
This argument also holds for the aperiodic word terminating the period-doubling cascade, which can also be proven to be maximal (as are all words in the cascade). That word forms the boundary between periodicity and chaos; a sequence of finite words of length 2^n ends in this word of length 2^{\aleph_0}, which then has the cardinality of \aleph_1. The periodic window theorem holds for the periodic 'penultimate word', and, since the ultimate word is maximal, the word resulting from replacing its final letter with a C is superstable. Continuity then implies a stable window connecting the penultimate word to the superstable point of the ultimate word, proving the existence of the left side of the window (for a plot in which the cascade on the left becomes chaos on the right). This is sufficient to prove there is a finite-measure window of stability in which the aperiodic word exists.
-
"The generalized composition rules are only stated as a sufficient criterion to produce a sequence of maximal words, which does not exclude that inflation rules which do not satisfy them can produce an infinite sequence of maximal words"
This is another good point. The purpose of Section 4.4, 'Other Time Quasilattices', is to address this issue. In that section I take as an example the Fibonacci quasilattice, and consider two possibilities. First, that the Fibonacci quasilattice might happen to form a stable time quasilattice even if its finite approximations are inadmissible. This case is uninteresting, since the infinite word cannot be tested in any finite-duration experiment. Second, that not only is the Fibonacci quasilattice admissible, but that there exist Fibonacci words of systematically increasing lengths which are also admissible. This is a harder case to dismiss. The Fibonacci inflation rules applied to the symbol R immediately lead to inadmissible words. However, generating a long section of the Fibonacci quasilattice and searching for admissible words of different lengths, the cumulative total number of admissible words increases logarithmically as a function of maximum word length sought (figure 7). The sub-linear growth hints that the Fibonacci quasilattice is inadmissible, but this is by no means a proof.
It is impossible to show that other time quasilattices do not exist, as one cannot prove a negative. All I can say is that I am unaware of another method by which they could be proved not only to exist, but to be reached by a systematic set of finite-length approximations of increasing length. It is the second criterion which makes time quasilattices identifiable in finite-duration experiments and therefore physically interesting.

---

## Round 3 · List of Changes

Warnings issued while processing user-supplied markup:
- Inconsistency: Markdown and reStructuredText syntaxes are mixed. Markdown will be used.
Add "#coerce:reST" or "#coerce:plain" as the first line of your text to force reStructuredText or no markup.
You may also contact the helpdesk if the formatting is incorrect and you are unable to edit your text.
-
A distinction is now drawn between time quasilattices and time quasicrystals. Most uses of 'time quasicrystal' now fall under the former case and have been rephrased, including in the title.
-
Line appended to Abstract: "We identify quantum many-body systems in which time quasilattices develop rigidity via the interaction of many degrees of freedom, thus constituting dissipative discrete 'time quasicrystals'."
-
Altered line in Introduction: "between these two extremes we find 'quasicrystals', aperiodic tilings consisting of two or more unit cells." --> "between these two extremes we find 'quasicrystals', atomic decorations of 'quasilattices' which are aperiodic tilings consisting of two or more unit cells."
-
Added paragraph to Introduction:
"Extending the crystal lattice analogy, we further identify time quasi\emph{crystals}: systems in which the symmetry of a periodic driving is spontaneously broken to the symmetry of a time quasilattice, in which the stability is made rigid by the interactions between the macroscopic number of degrees of freedom of a quantum many-body state. We examine a number of recent experimental proposals concerning discrete time crystals in driven dissipative many-body systems~\cite{GongEA18,WangEA18,YaoEA18}, identifying that several additionally host time quasicrystals. We detail experimental signatures of these new states of matter."
- Renamed section 2 "Quasilattices and Quasicrystals", and added these explanations of the nomenclature:
"Here we focus on the case of one-dimensional quasilattices, which can be generated as cuts through two-dimensional lattices. We reserve the name 'quasicrystal' for physical systems (quasilattice plus atomic basis) in dimensions two and higher. The phrase 'quasilattice' is used for the mathematical structure describing the physical system."
"In one dimension, it is a standard convention to refer to both the physical systems and their mathematical descriptions as quasilattices, disallowing the use of the term quasicrystal. This permits a precise definition of quasicrystals as those systems featuring diffraction patterns with symmetries forbidden by the crystallographic restriction theorem~\cite{SocolarSteinhardt86,Janot,Senechal,BoyleSteinhardt16,BoyleSteinhardt16B}. This definition precludes the possibility of quasicrystals in one dimension, as rotations are not well defined. The only break which we make with this convention is in Section~\ref{TQCs} in which we identify time quasilattices stabilized by many-body interactions: in order to emphasize that these states constitute an extension of the concept of time crystals to include quasilattice symmetry, we term them \emph{time quasicrystals}, despite the fact that they exist in one dimension of time."
- Added a new section, "Time Quasicrystals":
" In the previous sections we established the concept of time quasilattices: the mathematical structure of quasilattices, in the time direction. We found them as stable and structurally stable trajectories in dissipative dynamical systems. Until now we have not been concerned with the physical origin of the stabilising non-linearity, owing to the universality of chaotic dynamics~\cite{Strogatz}. After the present paper appeared online, a number of experimental proposals for realizing discrete time crystals in driven dissipative quantum many-body systems were proposed~\cite{GongEA18,WangEA18}. In this section we identify signatures of time quasilattices in these systems; as the structures are additionally rigid in the same sense as time crystals, we identify these responses as new states of matter, 'time quasicrystals'. We begin by providing precise definitions of these phrases before identifying signatures of the states.
Discrete time crystals feature a period-doubled response to a periodic driving~\cite{Sacha15,YaoEA17,ElseEA16,KhemaniEA16}. There should also be a sense of \emph{rigidity}, in order to bring them in line with our intuition regarding spatial crystals~\cite{YaoEA18}. The three cases of interest in the present context are as follows.
% [begin itemize environment] \item A \emph{time quasilattice} returns an aperiodic response to a periodic driving, featuring two unit cells of different durations, where each cell appears with precisely two spacings and the ratio of cell populations tends to a Pistot-Vijayaraghavan number as the number of cells tends to infinity. Both the durations of the cells and their sequence are stable and structurally stable to perturbations, so the order persists indefinitely. % \item A \emph{discrete time crystal} occurs when (i) the discrete time translation symmetry of a periodic driving is spontaneously broken by a lower-period response, which is (ii) made both stable and structurally stable to perturbations and finite temperature by (iii) the local interactions of many degrees of freedom, and which (iv) persists indefinitely. There should also be (v) a sense in which it can be understood to be a ground state. % \item A \emph{discrete time quasicrystal} occurs when the discrete time translation symmetry of a periodic driving is spontaneously broken by a time quasilattice response, which is made both stable and structurally stable to perturbations and finite temperature by the local interactions of many degrees of freedom, and which persists indefinitely. There should also be a sense in which it can be understood to be a ground state. [end itemize environment] %
Note that this definition of time crystals does not necessarily include quantum mechanical effects; classical discrete time crystals have been proposed, and the original proposal for classical time crystals, which break continuous time translation symetry, was not ruled out by the no-go theorems applied to the quantum case~\cite{YaoEA18,WilczekShapere12,Watanabe15}.
Some leeway is built into requirement (v), since the concept of a true ground state requires energy to be conserved, which is not the case in any of the known examples of discrete time crystals. In refs.~\cite{Sacha15,YaoEA17,ElseEA16,KhemaniEA16} the periodic driving leads to a pseudo-energy being conserved modulo $2\pi$, and it is in this sense requirement (v) is fulfilled. Reference~\cite{YaoEA18} uses the phrase \emph{rigid subharmonic entrainment} for dissipative systems fulfilling the other criteria, reserving the phrase \emph{classical discrete time crystals} for the case in which the classical many-body system remains rigid when coupled to a finite-temperature bath (although, since inherently out-of-equilibrium, the concept of a ground state is again avoided). Other references refer to these states as \emph{dissipative discrete time crystals}~\cite{GongEA18} or equivalent phrases~\cite{WangEA18,RussomannoEA17}. This is the convention we adopt here.
The advantage of explicitly allowing dissipation is that states beyond period doubling can be stabilised. In reference~\cite{GongEA18} a protocol is outlined to identify dissipative discrete time crystals in quantum many-body cavity/circuit QED setups governed by the Dicke model. Numerical simulations of the classical limit show several signatures the authors identify in a simplified discrete-time nonlinear model featuring a period-doubling cascade into chaos. The authors further argue that these signatures are also present in the quantum many-body limit which would be realised by the experiments they propose. The experimental identification of a period-doubling cascade is a sufficient condition for all of the Pell and Clapeyron words to appear, and would prove that these systems feature time quasilattices. Since the stability derives from the interactions of many degrees of freedom, and the quantum many-body system features spontaneous symmetry breaking into this state, these setups would then feature true (dissipative, discrete) \emph{time quasicrystals}.
In reference~\cite{WangEA18} dissipative discrete time crystals are identified in a numerical model of a driven open quantum system (bosonic atoms in a double-well potential). There is a clear period doubling cascade into chaos in the model's classical limit -- again, a sufficient condition for the presence of time quasilattices, and therefore in this scenario (dissipative, discrete) time quasicrystals. The authors also identify the continuation of the classical period-doubled state to the quantum regime via two-time correlation functions. This opens the possibility of identifying \emph{quantum} dissipative discrete time quasicrystals in this system.
A proposal for a non-dissipative discrete time crystal based on a kicked Lipkin-Gleshkov-Glick model is provided in~\cite{RussomannoEA17}. This model features a Hamiltonian system of spins with a periodic driving, and the authors identify candidate experimental implementations in Bose Einstein condensates and trapped ion systems. Rigid responses of various periods are identified within a classically chaotic regime, although no period-doubling cascade is immediately obvious. Strictly, the proposals in both references~\cite{GongEA18} and \cite{RussomannoEA17} violate requirement (iii) above, since the many-body interactions stabilizing the discrete time crystal phases are infinite-range rather than local. Local couplings ensure that the concept of dimensionality is well-defined in abstract mathematical models: depending on the topology of local connections, a model of many-body interactions could correspond to a range of physical dimensions. In the present context of physical interacting particles, however, this requirement seems unnecessarily limiting (the systems are all three dimensional).
In all these cases, the procedure for identifying the time quasicrystals in the classical regime would be to identify the sequence of periodic approximations (the finite-length Pell or Clapeyron words) as an externally-tunable field is varied. The external field depends on the individual systems~\cite{GongEA18,WangEA18,RussomannoEA17}. Several Pell and Clapeyron words can already be seen without further analysis in the classical limit of reference~\cite{WangEA18}. The time quasicrystals themselves are indistinguishable from any of their periodic approximants featuring a period longer than the observation time. Nevertheless, their existences and stabilities are guaranteed by the periodic window theorem~\cite{HaoEA83}.
In experimental searches for time quasicrystals, the only things which can be measured are periodic approximations. This restriction is made necessary by the finite duration of the experiment. Each periodic response is simply a dissipative discrete time crystal, and so the techniques developed in the references already suffice to identify them in both the classical and quantum regimes. The only extension necessary experimentally would be to identify that, as a function of the tunable system parameters, a sequence of time crystals is found with periods increasing as either the Pell or Clapeyron words. "
-
Added references to the new section throughout the manuscript, as appropriate.
-
Added comment to Conclusions:
"Interestingly, chaotic systems can demonstrate synchronization when coupled, while maintaining their unpredictability~\cite{Strogatz,Gonzalez,ArenasEA08}. This synchronization can take the form of a fixed delay between points on the particles' trajectories; [added text:] it has even been demonstrated to persist to the quantum regime of systems with a chaotic classical limit~\cite{LeeEA13,HushEA15,LorchEA17}."
- Added paragraph to Conclusions:
"Several experimental proposals have recently appeared for discrete time crystals in dissipative systems~\cite{GongEA18,WangEA18}. Numerical simulations show clear examples of period-doubling cascades into chaos in the systems' classical limits. This is a sufficient condition for realizing time quasilattices. If identified experimentally these systems would therefore feature (dissipative, discrete) \emph{time quasicrystals}: the discrete time translation symmetry of a periodic driving is spontaneously broken to the symmetry of a time quasilattice, which is stabilised against perturbations via the local interactions of a quantum many-body state. They can be experimentally identified by their sequences of periodic approximations (finite Pell and Clapeyron words)."
- minor rephrasings or re-orderings of existing text to accommodate the listed changes.

---

## Editorial Decision

published